# Micro-scale functional modules in the human temporal lobe

Julio I. Chapeton [1]✉, John H. Wittig Jr[1], Sara K. Inati [1] &
Kareem A. Zaghloul [1]✉

The sensory cortices of many mammals are often organized into modules in the form of cortical columns, yet whether modular organization at this spatial scale is a general property of the human neocortex is unknown. The strongest evidence for modularity arises when measures of connectivity, structure, and function converge. Here we use microelectrode recordings in humans to examine functional connectivity and neuronal spiking responses in order to assess modularity in submillimeter scale networks. We find that the human temporal lobe consists of temporally persistent spatially compact modules approximately 1.3mm in diameter. Functionally, the information coded by single neurons during an image categorization task is more similar for neurons belonging to the same module than for neurons from different modules. The geometry, connectivity, and spiking responses of these local cortical networks provide converging evidence that the human temporal lobe is organized into functional modules at the micro scale.

The primary sensory cortices of many mammals can often be divided into distinct cortical columns containing neurons with similar tuning properties[1,2]. This type of organization falls under the broader concept of modular networks, where a modular network is one that can be naturally partitioned into groups of nodes, or modules, that have denser connectivity to each other than to the rest of the network[3–7]. Networks comprised of spatially compact modules possess fundamental evolutionary[8–12], metabolic[13–15], and computational advantages[16–20]. Such modular networks that can produce complex dynamics while being metabolically efficient may be particularly relevant for human cognition[21,22].

The clearest examples of cortical modules in which measures of connectivity, structure, and function converge are the barrel columns in the rodent somatosensory cortex. These columns can be visualized with a variety of histology methods, and neurons within each column share similar tuning properties and exhibit denser connectivity to one another than to neurons from other columns[2,23–27]. In non-human primates, systematic investigations of inferior temporal cortex have established the presence of columnar modules in which cells within the same column respond to similar though not necessarily identical features, while cells in different columns respond to different features[28,29]. Models using simple iterated functional modules can

produce object representations that approximate the neuronal responses of these cells and perform reasonably well in object recognition tasks when compared to humans[30]. These findings coupled with the aforementioned theoretical considerations have motivated the hypothesis that modular organization in the form of cortical columns may in fact be a general principle for cortical circuits.

In humans, however, clear examples of modular organization at the scale of cortical columns are restricted to the primary visual cortex, in which anatomical and high magnetic field studies have revealed maps of retinotopy, ocular dominance, and orientation selectivity[31–39]. These studies rely upon the fact that the response properties of the primary visual cortex are well known, and stimulus contrasts can be designed to strongly and differentially activate local patches of tissue. The response properties of local circuits in higher-order brain regions are less clear, however, limiting the ability to functionally map the human brain at the submillimeter scale. Functional imaging and electrocorticography studies have identified function specific modules in higher-order brain regions, but only at larger spatial scales and in larger brain patches. Some examples include the face selective fusiform face area[40–43] and the intraparietal sulcus region in the case of numerical processing[44,45]. But there has been no direct evidence that higher-order regions in the human brain

[1]Surgical Neurology Branch, NINDS, National Institutes of Health, Bethesda, MD 20892, USA. ✉e-mail: julio.chapeton@nih.gov; kareem.zaghloul@nih.gov

are organized into functional modules at the spatial scale of cortical columns.

Here, we test the hypothesis that the human temporal association cortex, a higher-order brain region involved in semantic processing, exhibits a modular organization at the micro scale. We tested this hypothesis by specifically investigating whether the functional circuit of this brain region displays the following features: (i) If the temporal lobe is organized into modules at this spatial scale, then these modules should have dense internal connectivity and sparser connectivity to neighboring modules. (ii) Similar to the functional columns identified in the primary visual cortex, cortical modules in the human temporal lobe should be spatially contiguous ellipsoids around 1mm in width, and (iii) if they are constrained by slowly-changing structural connectivity they should also be reproducibly identified from day to day. (iv) The information coded by neurons within a module should be more similar than information coded by neurons from different modules, and (v) the boundaries between modules should be sharp in terms of both functional connectivity and neural responses.

To assess if local temporal lobe circuits exhibit these features, we analyzed local field potential (LFP) and single unit spiking activity captured through microelectrode arrays (MEAs) implanted in participants with drug resistant epilepsy who were being

monitored for seizures. Based on the LFPs captured in each MEA, we constructed weighted and directed functional networks, and analyzed the evolution of these networks over several days. We find that local cortical networks in the temporal lobe are partitioned into spatially compact modules approximately 1.3 mm in width, and that these modules persist over time and different cognitive demands. Critically, in response to images presented during a category recognition task, neurons within modules have more correlated neural responses and share more information regarding the semantic category of the image than neurons from different modules. The connectivity, spatial geometry, and functional properties of these networks support the hypothesis that the modular organization that has long been observed in primary sensory cortices may also characterize the micro-scale circuitry of the human temporal lobe.

## Results

### Directed functional connectivity in local circuits of the temporal lobe

We analyzed local field potentials (LFPs) captured from microelectrode arrays (MEAs) implanted in the anterior middle temporal gyrus of eight participants who were being monitored with intracranial

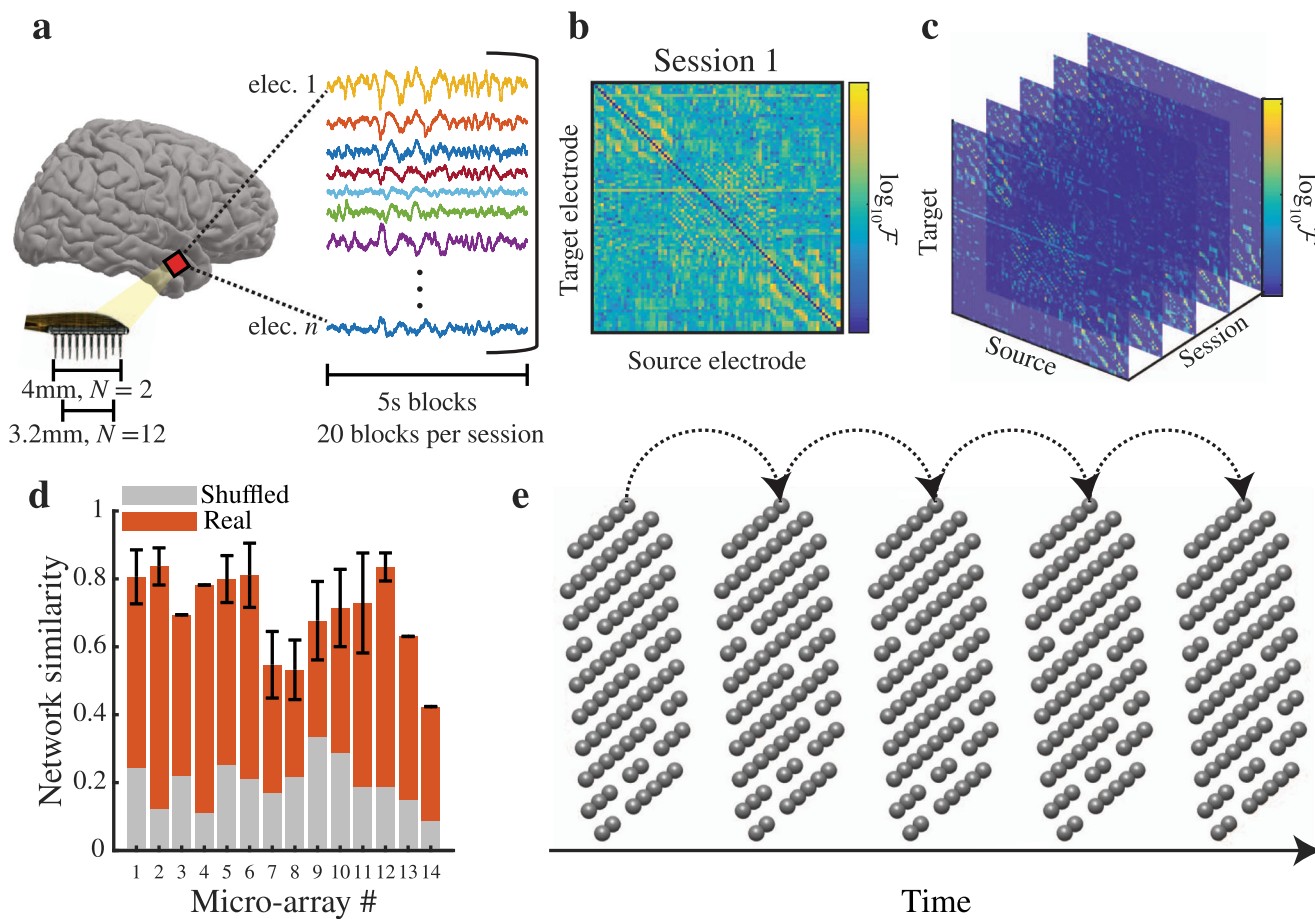

**Fig. 1 | Directed functional connectivity in local cortical networks. a** Schematic of recordings and individual blocks. From each MEA we extract 20 data blocks, 5 s each, for functional connectivity analysis. **b** Weighted and directed adjacency matrix for a single session. For each session we construct a weighted and directed adjacency matrix by computing the pair-wise conditional Granger causality, $\mathcal{F}$, between every pair of electrodes. For ease of visualization the colormap shows $\log_{10}\mathcal{F}$. **c** For each individual MEA we threshold the adjacency matrix from each session and set all values that were not significantly different from zero equal to zero. The end result is a series of adjacency matrices, or layers, where each layer

corresponds to a session. **d** Adjacency matrix similarity across sessions. For every MEA, the average correlation between adjacency matrices from different sessions (orange bars) is large ($0.4 < r < 0.9$) and larger than the maximum similarity between pairs of shuffled networks (gray bars, $n = 500$ surrogates). **e** Multi-layer network representation. By attaching links between the same node across layers, we can represent adjacency matrices from all sessions as a single multi-layer network (dotted lines show the intra-layer links for a single node). This allows us to find the optimal partition of the network into communities for all sessions at once.

electrodes for drug resistant epilepsy. We used a single MEA in two of the participants (4 mm × 4 mm with 96 electrodes), and two smaller MEAs in the remaining participants (3.2 mm × 3.2 mm with 64 electrodes each). We considered each of the 14 total MEAs separately (implantation sites shown in Supplementary Fig. 1a). Participants performed cognitive tasks during several different experimental sessions throughout the monitoring period, and we extracted 20 separate blocks of data (each of duration 5 s) from each experimental session for connectivity analyses (Fig. 1a; Supplementary Fig. 1b, c; see Methods).

In every experimental session, we computed the pair-wise conditional Granger causality (GC), $\mathcal{F}_{j \to i}$, between all pairs of microelectrodes to construct a weighted-directed adjacency matrix of functional connections for each MEA (Fig. 1b). Pair-wise conditional GC quantifies the degree to which activity of a source electrode can predict the future activity of a second target electrode over and above the degree to which the target electrode's activity is already predicted by its own past and the past of all other electrodes besides the source[46]. This helps mitigate spurious causalities that may arise due to common dependencies or instantaneous field effects like volume conduction[47]. Moreover, Granger causal methods have shown a higher correspondence to anatomical connectivity compared to other measures of functional connectivity[48]. For each MEA in each session we generated a weighted and directed network representation by treating each microelectrode as a node and the GC values as the connection weights between the nodes. We set entries in the adjacency matrix that are not significantly different from zero to be exactly zero (Fig. 1c; see Methods)[49].

Across all MEAs, we find that local functional networks in the temporal lobe are sparse (connection probability = 0.26 ± 0.08), and the functional network in each individual MEA is highly similar across experimental sessions. For each MEA, we quantified this similarity by computing the average correlation between every pair of adjacency matrices from different experimental sessions (mean correlation = 0.7 ± 0.13 across MEAs; Fig. 1d). We compared these correlations to the maximum similarity that would be observed between randomized networks with the same degree distributions as the real networks ($n$ = 500 surrogates for each MEA; see Methods)[50]. In each MEA, the similarity of the network between every pair of experimental sessions is significantly greater than the maximum similarity between the surrogate networks, demonstrating that the strength and direction of the identified functional connections are stable across several days.

## Local functional networks are composed of temporally persistent modules

The stability of micro-scale functional networks motivated us to use a multilayer modularity approach that allows us to identify and track the evolution of network modules across experimental sessions. We considered each adjacency matrix from each session as an individual layer, stacked the layers chronologically, and connected each node to itself across consecutive layers to generate a single multilayer network for each MEA (Fig. 1e; see Methods). This type of multilayer network is also known as a temporal network[51,52]. We then used multilayer modularity maximization to find the best partition of the network into communities across all sessions[53,54]. This approach optimizes a multilayer quality function, $Q_{ml}$, which is bounded from above by 1 and quantifies how well a partition concentrates connectivity within modules. In practice, values larger than 0.3 are taken as evidence for a modular network[55]. One of the main advantages of the multilayer approach is that communities maintain their labels across the layers so that individual communities can be tracked over time. The end result of optimizing the multilayer modularity function is a time-dependent partition of nodes into modules.

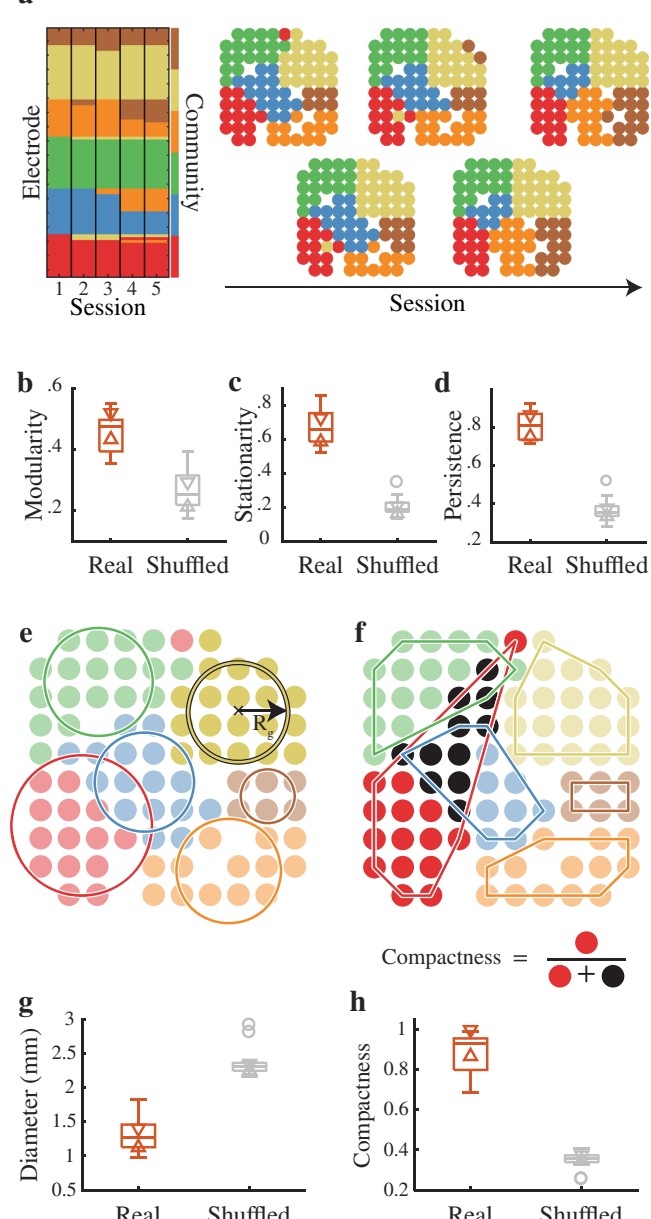

Local networks in the temporal lobe are highly modular ($Q_{ml} > 0.35$ for every MEA), and membership in each module does not change substantially from one session to the next (Fig. 2a; Supplementary Fig. 1d; Supplementary Fig. 2 for all MEAs). Although we used the pair-wise conditional GC to define functional connectivity between nodes, the stronger connectivity within modules and weaker connectivity across modules is evident even when using other metrics of connectivity such as correlation or coherence (Supplementary Fig. 3). Because optimizing modularity will always partition a network into individual modules, it is important to determine if the quality of this partition is higher than what would be expected by chance from random networks with similar statistics. Hence, for each MEA we compared the modularity to the modularity that would arise in random networks with the same degree distribution as the real networks ($n$ = 500 surrogates for each MEA). Local networks in the temporal lobe are significantly more modular than what would be expected by chance both in each individual MEA (Supplementary Fig. 4a; Supplementary Table 1) as well as across all MEAs ($Q_{ml} = 0.45 ± 0.06$ vs

**Fig. 2 | Quality, temporal evolution, and geometry of network partitions. a** (*left*) Optimal multilayer partition for an individual MEA. Every row represents an electrode and every column a session. The color of each cell denotes the community that node was assigned to for that session. Electrodes were sorted according to their community assignment for the first session. (*right*) Spatial embedding of individual modules. The locations of the electrodes are shown with each electrode colored according to the community it was assigned to. **b** Modularity index across all MEAs. Across all MEAs the modularity of the real networks is significantly larger than for degree-matched random networks. **c** The stationarity across all MEAs is significantly larger than for degree-matched random networks. **d** The persistence across all MEAs is significantly larger than for degree-matched random networks. **e** Illustration of module spatial extent. The module diameter is estimated as $2R_g$, where $R_g$ is the radius of gyration for a set of equal masses. **f** Illustration of module compactness. The module compactness is the number of nodes from a module that are within the module's convex hull divided by the total number of nodes within the hull. **g** Module spatial extent across all MEAs. The module diameter is consistent across all MEAs ($1.31 \pm 0.24$ mm). For reference, the average module diameter for degree-matched random networks is closer to the size of the entire MEA ($\approx 2$–$3$ mm). **h** The mean compactness is near 1 and significantly larger than for degree-matched random networks, indicating that the modules are mostly contiguous in space. For all box-plots the whiskers cover points within 1.5 IQR (interquartile range) of the first or third quartile; values outside of this range are considered outliers and are represented by individual circles. The notches (triangles) are placed at the median $\pm 1.57 \frac{IQR}{\sqrt{n}}$, where $n$ is the number of MEAs.

$0.27 \pm 0.06$ for the surrogates; $t(13) = 18.16$, $p < 0.001$; paired $t$-test; Fig. 2b).

To test how reproducible these partitions are from day to day we computed the stationarity and normalized persistence of the multilayer partitions[54,56,57]. Stationarity quantifies how quickly or slowly community structure evolves from one time point to the next and is defined as the average correlation between subsequent partitions for each MEA. The partitions of each local network we examined in the human temporal lobe change slowly from one session to the next, and change significantly more slowly than expected by chance (Supplementary Fig. 4b; Supplementary Table 1). Across all MEAs together, we found that the average stationarity of the partitions is significantly greater than the stationarity expected by chance ($0.67 \pm 0.11$ vs $0.20 \pm 0.06$; $t(13) = 15.37$, $p < 0.001$; paired $t$-test; Fig. 2c). Normalized persistence tallies the number of nodes that do not change modules between sessions, normalized by the total number of nodes. We found that within each MEA, the vast majority of nodes remain within the same module across all sessions ($0.87 \pm 0.04$ for the example MEA; Supplementary Fig. 4c; Supplementary Table 1). Across all MEAs, the average persistence is significantly greater than the persistence expected by chance ($0.81 \pm 0.07$ vs $0.37 \pm 0.06$; $t(13) = 19.42$, $p < 0.001$; paired $t$-test; Fig. 2d). The stability in modular organization observed in these local functional networks raises the possibility that these partitions arise from the hypothesized columnar organization of local structural networks in the cortex.

## Spatial geometry of modular networks

To examine if the geometry of these modules is consistent with columnar organization, we used the root mean square distance from all nodes within a module to the module's centroid to generate an estimate of each module's diameter (Fig. 2e; see Methods). In each MEA, the distribution of module diameters across all modules and all experimental sessions exhibits a stereotypical size for each individual module of approximately 1.3 mm, similar to the size of the ocular dominance columns in the human visual cortex[31–34,36–38] (Supplementary Fig. 5a for all MEAs). In each case, the distribution of module diameters is significantly smaller than the module diameters that are present in the surrogate networks (Supplementary Table 2). On average across all MEAs, the identified modules are significantly smaller than the modules that emerge from the surrogate networks which, due to the lack of any spatial embedding, tend to have sizes closer to the

size of the entire array ($1.31 \pm 0.24$ mm vs $2.37 \pm 0.22$ mm; $t(13) = -17.46$, $p < 0.001$; Fig. 2g).

We used a metric of spatial compactness to quantify how contiguous a module is in space (Fig. 2f; see Methods). This metric takes on a maximum value of 1 if all nodes within a module are spatially contiguous. In each individual MEA, we found that the modules are indeed spatially compact (Supplementary Fig. 5b), and more compact than what would be expected by chance (Supplementary Table 2). On average, across all MEAs the spatial compactness of the true local networks is significantly greater than the spatial compactness of the random networks ($0.89 \pm 0.1$ vs $0.35 \pm 0.05$; $t(13) = 20.97$, $p < 0.001$; Fig. 2h).

We did not find any systematic differences in geometry due to the different array sizes (3.2 mm vs 4 mm; Supplementary Fig. 5c). In addition, the quality, persistence, and size of the modules we identified are relatively robust to different parameters used to maximize modularity (see Supplementary Fig. 6), supporting the claim that modules in this patch of cortex have a stereotypical size and shape.

## Frequency specific contributions to within vs across module connectivity

Oscillations at different frequencies have been proposed as a mechanism to coordinate neural activity across multiple spatial and temporal scales[48,58–61]. Modular networks can produce faster processes within modules and slower processes across modules, offering a natural way for the brain to process information locally while also integrating information from spatially distributed neuronal populations[17,62]. To test for this possibility, we calculated the relative contribution of each individual frequency to the total Granger causality for each connection (see Methods). We separated the connections into those that are contained within a module and those that span across modules. On average, within-module connectivity is stronger at all frequencies, consistent with the definition of modularity (Fig. 3a for an individual MEA). We can account for these overall shifts in the GC spectrum by normalizing the individual spectra, allowing us to compare the relative frequency contributions to connectivity within and across modules (Fig. 3b).

Within individual MEAs, we found that the average normalized GC spectrum across sessions for within-module electrode pairs has a larger contribution from higher frequencies compared to across-module electrode pairs (Fig. 3b inset). To quantify this effect across MEAs we aggregated frequencies into five frequency bands: delta (2-4Hz), theta (4-8Hz), alpha (8-16Hz), beta (16-32Hz), and broadband gamma (70-150). Across all MEAs, functional connectivity within modules is mediated by significantly greater high frequency activity and significantly less low frequency activity than connectivity across modules (Fig. 3c).

## Differential coding by functional modules

Although functional connectivity suggests that local networks in the human temporal lobe are organized into spatially compact modules, establishing that the identified modules are indeed functionally distinct from one another requires evidence that neurons within different modules respond differentially during cognitive processing. A subset of participants performed an image categorization task while we recorded single unit spiking activity from a population of neurons using each MEA (5 participants, $n = 10$ MEAs; 10 total experimental sessions; Fig. 4a; Supplementary Fig. 7; see Methods). Because the arrays were implanted in a region often implicated in processing semantic representations, particularly of famous people and places[63–65], we tested whether the spiking responses could be used to discriminate image categories, and if so, whether such neural coding differed between neurons belonging to different modules. We analyzed each session from each MEA independently since the set of units recorded from the same MEA on one day is almost entirely different from the set of units recorded on a separate day[66].

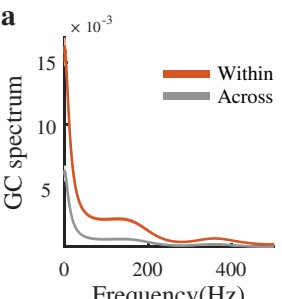
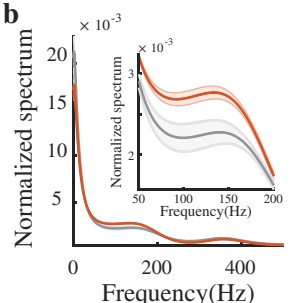
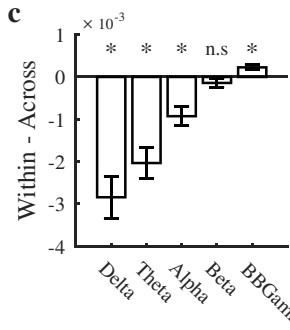

**Fig. 3 | Within- vs across-module communication. a** Average raw Granger spectra for an individual MEA. **b** Normalizing the Granger causality spectra to have area 1 allows for the comparison of within- vs across-module connectivity as a function of frequency. **c** Difference in GC as a function of frequency band across all MEAs. Across all MEAs, within-module communication contains significantly larger broadband gamma and significantly smaller delta, theta, and alpha contributions when compared to across-module communication. There is no significant difference for the beta band.

In an example session from an individual MEA, neuronal responses to an image of a famous person appear to depend on module membership (Fig. 4b). The magnitude and timing of spiking responses appear consistent within modules, and more heterogeneous across different modules. This was true for this individual trial, as well as on average across all trials for this behavioral session. To quantify this, we trained a linear classifier to discriminate among image categories using only the spiking neurons within each identified module (median 23 neurons per module, range 7 to 34 for the example session and MEA). In this example, by 500 ms after stimulus presentation, classification accuracy rises above chance levels in five of the seven identified modules (Fig. 4c; see Methods). In 26 of the 75 modules identified across all sessions and MEAs, overall classification accuracy using the population spiking activity in that module is significantly greater than chance (Supplementary Fig. 8a). Accuracy tends to be higher for modules that contain more neurons, although this relation is not absolute (Spearman's $\rho = 0.51$, $p < 0.001$, Supplementary Fig. 8c).

The observed heterogeneity in classification accuracy across modules, even within the same MEA, supports the hypothesis that the modules are functionally distinct. It is possible, however, that a random and spatially distributed subset of individual neurons code image category, and that modules containing more neurons are more likely to include those image category neurons simply by chance. If in fact the modules do define patches of cortex that share similar functional responses, then neurons within each module should share more information about each image than neurons from different modules. We directly tested for this by comparing classification accuracy after shuffling neuron identities in two ways. In the first, for each module we shuffled the neuron identities using only neurons from that individual module before classifying the hold-out set, whereas in the second, we shuffled neuron identities with neurons from an entirely different module (Fig. 4d; see Methods). In an example module, shuffling neuron identity within the module decreases classification accuracy, but overall classification still remains high and significantly greater than chance ($p < 0.001$, $t(19) = 9.61$ paired $t$-test; Fig. 4e). In contrast, when shuffling neuron identities with neurons from a different module with comparable classification accuracy, classification accuracy drops substantially and hovers near the chance level. Across all modules in this MEA and session, classification accuracy is significantly better after shuffling within compared to shuffling across modules ($t(41) = 5.2$, $p < 0.001$; Fig. 4f). This difference is consistent across all modules from all MEAs and behavioral sessions ($t(297) = 9.7$, $p < 0.001$; Fig. 4g), and across sessions (module-averaged difference per session and MEA $t(15) = 3.1$, $p = 0.007$). The heterogeneity in coding within small patches of tissue together with the shuffling analysis shows that the modules are functionally distinct. Neurons belonging to the same module code similar, though not identical, information regarding image category,

whereas category information is more independent across neurons belonging to different modules.

## Module boundaries are sharp

Within the context of brain mapping a distinction can be made between modules, where functional boundaries are sharp, and functional maps, where stimulus selectivity changes gradually[41]. We therefore analyzed both the functional connectivity and similarity in neural responses between electrodes at the boundaries of each module. We defined the boundary for each module using it's convex hull, and identified electrodes on the boundary that have a nearest neighbor that is inside the boundary as well as a nearest neighbor that is outside of the boundary (Fig. 5a). In this manner, we were able to compare connectivity within and across modules while exactly matching for distance. Across all MEAs, the average GC over all modules is larger for comparisons within the boundary versus across ($t(13) = 5.4$, $p < 0.001$; Fig. 5b).

We performed an analogous analysis using the average correlation across all trials between neuronal spiking responses captured during the categorization task (Fig. 5c). Because some units may not fire at all during most trials, we restricted this analysis only to units that spiked in at least half of the trials. Unlike the LFPs which are robustly captured from nearly every electrode, the spatial coverage with respect the single unit activity is much sparser. Nevertheless, across all available comparisons ($n = 21$ modules from 9 sessions and 5 MEAs) we find that responses for pairs within the boundary are significantly and substantially more correlated than for pairs across the boundary ($t(20) = 3.4$, $p = 0.003$; Fig. 5d). In these analyses comparisons between within module and across module pairs are exactly matched for distance, highlighting that the boundaries of these partitions are not arbitrary.

## Discussion

In this study we provide evidence that local circuits in the human temporal lobe are organized into spatially compact functional modules. These modules, identified by partitioning networks constructed using measures of directed functional connectivity at the submillimeter scale, persist over time and have a diameter of approximately 1.3 mm. Importantly, these modules have functional significance. During an image categorization task, neurons belonging to the same module share more information about visual stimuli than those belonging to different modules, and the boundaries of these modules are sharp in terms of both functional connectivity and neural responses.

In complex networks, modularity is typically defined based on measures of connectivity, i.e., a modular network is one in which there is strong connectivity within modules and weaker connectivity across

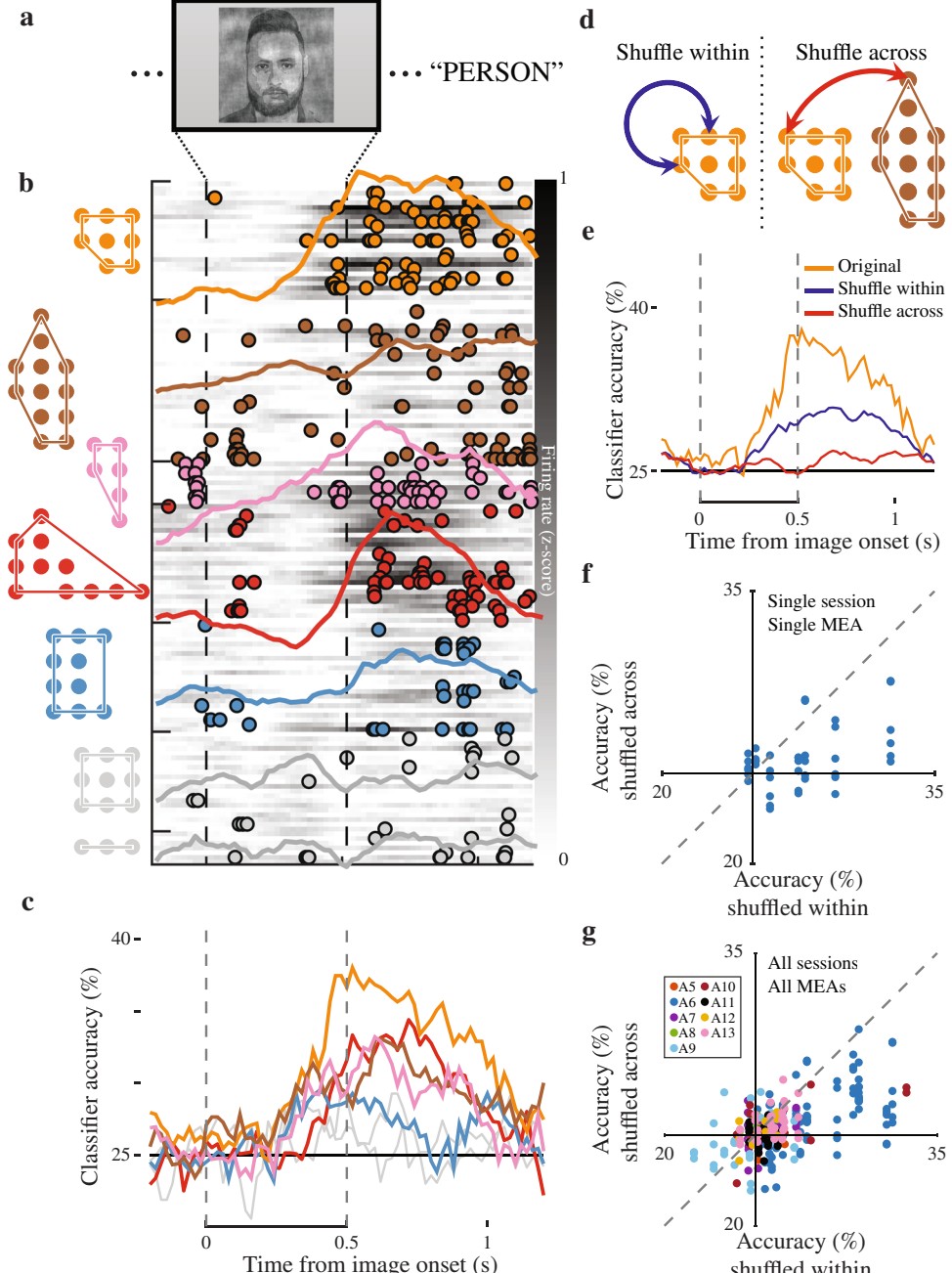

**Fig. 4 | Differential category coding by individual modules. a** Category recognition task. Participants are shown images on a laptop screen and asked to categorize them as people, places, animals, or objects. During the 'PERSON' trials participants are shown pictures of famous people; due to licensing requirements we show a representative image that is not part of the stimulus set. **b** Response properties of individual modules. For an individual 'PERSON' trial the spike times are colored according to the module that the spike came from (left). The grayscale background represents the trial-averaged firing rate for each individual unit, and the colored lines show the average firing rate for each module. Neuronal spiking appears to cluster in time according to module membership. This was true for individual trials as well as on average across all 'PERSON' trials. **c** Coding of image category by individual modules. Each line represents the time-course of classification when using data from only that module. Some modules contain information about the category of the image (colored lines) whereas classification for others was below chance level (gray lines). **d** Schematic of within vs across module coding analysis. For the shuffle within, we shuffle unit identities using only units from that individual module before classifying a hold-out set, whereas for the shuffle across we shuffle the data only from units that belong to different modules. **e** Effects of shuffling on category classification. In this example, accuracy decreased when shuffling within, but remained well above chance level. In contrast, after shuffling across the classifier accuracy drops to near chance. **f** Scatter of classifier accuracy after shuffling within and after shuffling across for a single session. The shuffle within accuracy was significantly higher than the shuffle across. **g** Scatter of shuffle within and shuffle across accuracy for all sessions. Across all sessions, shuffle within accuracy is significantly higher than the shuffle across accuracy. Different colored points denote the individual arrays.

modules. From the perspective of systems neuroscience, however, evidence for modularity often relies upon functional differences between modules. When activated by external stimuli, different modules should exhibit different response properties and tuning. In

this context, identifying cortical modules in primary sensory cortices, and specifically assigning them functional significance, is tractable because there are often well defined mappings between sensory inputs and neural responses. Still, the strongest evidence for modular

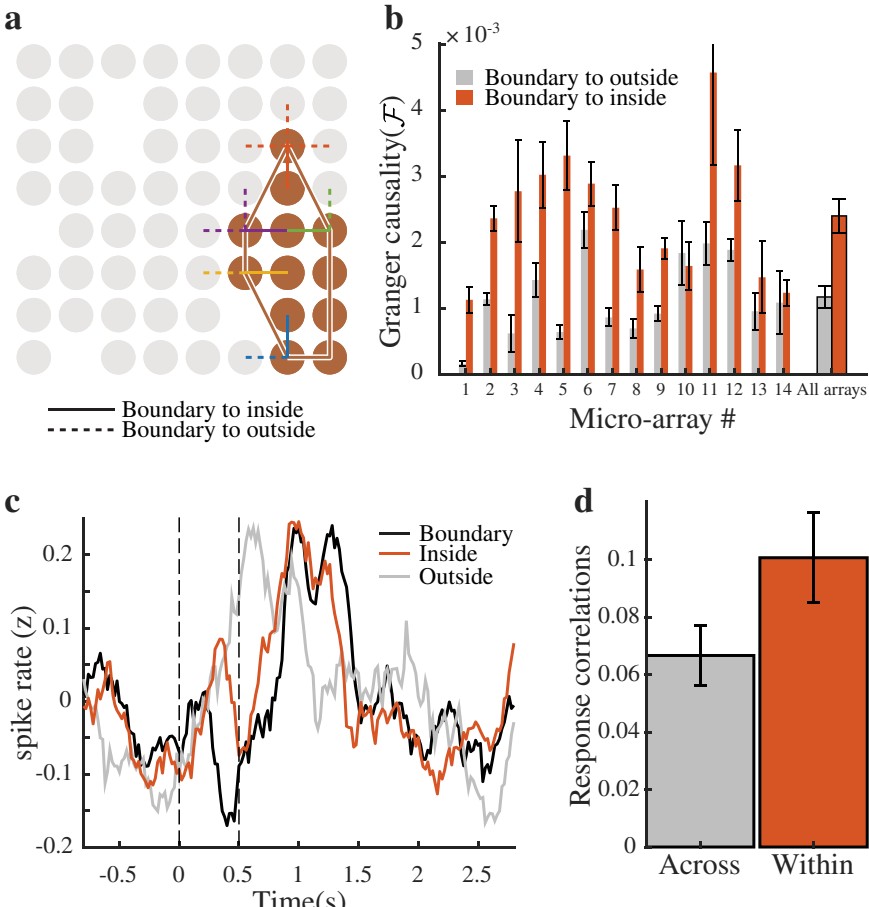

**Fig. 5 | Module boundaries are sharp. a** We use a partition's convex hull to define its boundary, and then identify electrodes on the boundary that have a nearest neighbor that is inside the boundary and a nearest neighbor that is outside of the boundary (solid vs dashed lines). **b** Mean pairwise conditional Granger causality for boundary to inside and boundary to outside pairs for the individual MEAs and across all MEAs (last pair of bars). **c** Trial-averaged responses during the category recognition task from a unit on the boundary, a nearest neighbor inside the boundary, and a nearest neighbor outside of the boundary. **d** The average correlation across trials between neural responses (or response similarity) for within and across boundary pairs. Error bars in (**b**) and (**d**) represent standard errors.

organization in neural circuits arises when measures based on connectivity, structure, and function converge, as is the case with the cortical columns of the rodent barrel cortex[23,24,27].

A central challenge in identifying modular organization in higher-order regions of the human brain, however, has been establishing convergence between connectivity and function in micro-scale networks. Obtaining precise knowledge of the connectivity diagram at the submillimeter scale required to probe cortical columns is rare. Therefore, functional modules in humans are almost exclusively defined using stimulus contrasts. Several neuroimaging and electrocorticography studies have examined stimulus responses to identify functional modules at larger spatial scales and in larger patches of cortex, such as the fusiform face area[40–43] and the intraparietal sulcus region[44,45], but examples of modular organization at the micro scale in the human brain are mostly restricted to the primary visual cortex[31–34,36–39]. In higher-order brain regions, the preferred stimuli for evoking localized neural responses at the micro scale are less clear. Coupled with the limitations in mapping connectivity at this spatial scale, obtaining convergence between connectivity and function to identify micro-scale modules in these regions has therefore been more difficult.

The lattice arrangement of the microarrays used in this study provides a convenient template for studying the spatial properties of local connectivity. Ideally, one would like to define connectivity based on interactions between the activity of single neurons. However, high-quality single unit activity of the type required for detailed analysis of spike-spike interactions is typically only available in a subset of recordings and from a subset of channels in the microarray, limiting the spatial interactions that can be assessed. Our approach addresses this challenge by using measures of functional connectivity to first identify modules from LFP signals, and then independently examining spiking activity of neurons within those identified modules to confirm the functional differences between them. While our results do not demonstrate explicit structural connectivity, the convergence of network functional connectivity and neural responses in these local circuits provides evidence that this higher-order brain region exhibits a modular organization. This strategy of first partitioning functional networks constructed from LFPs and then separately validating that these modules exhibit independent functional properties based on the spiking responses of individual neurons, may provide a generalizable approach for identifying the boundaries of functional modules throughout the cortex. These partitions can then be used to guide subsequent analyses.

A convenient consequence of this approach is that it allows us to also examine connectivity as a function of LFP frequency. We find a consistent difference in frequency content between connections within modules and connections across different modules. In relative terms, communication within modules is mediated by higher frequency components, while communication across modules is

mediated by lower frequency components. This asymmetry in frequency content may be related to oscillatory mechanisms of cortico-cortical communication[67–70]. Indeed, there is evidence to suggest that the specific oscillatory frequency involved in cortico-cortical communication may be related to the spatial scale over which communication occurs[58,61,71], consistent with our observations.

Processes like neural synchronization and cortico-cortical communication may operate over multiple spatio-temporal scales; similarly, cortical modules span multiple scales in a hierarchical fashion[21]. Probing human brain networks across multiple spatial scales is therefore critical for understanding how modular organization can subserve function. Unfortunately, no single methodology currently exists that can adequately sample activity at all of these spatial scales simultaneously. While the MEAs used here are well suited for probing local structure, there is a limitation on the extent of spatial coverage, particularly when working with human patients in a clinical setting. Studies using high-density electrocorticography, which offers broader spatial coverage at the expense of spatial resolution, have previously demonstrated that language processing in the temporal lobe is not spatially homogeneous at a separation of 4 mm[72]. Combining high-density electrocorticography with MEAs or studies with multiple contiguous MEAs will be required to link together these different spatial scales and determine the extent and scale of hierarchical modularity in human brain networks.

The geometry of the temporal lobe modules we identify, spatially contiguous ellipsoids approximately 1.3 mm in diameter, suggests that these modules could in fact reflect cortical columns. Early work in humans using cytochrome oxidase staining identified ocular dominance columns with similar dimensions[31,32,37], as have more recent neuroimaging studies[33,34,36,38]. Our results suggest that this columnar organization may be preserved in the human anterior temporal lobe, although recordings from multiple cortical layers will be required to fully asses the columnarity of these circuits.

While the functional relevance of columnar organization has been debated[73], the fact remains that the clustering of neurons with similar response properties into spatially segregated cortical columns is a ubiquitous feature of sensory cortices across various species, including tonotopic maps in the auditory cortex, taste maps in the gustatory cortex, somatotopic maps in somatosensory areas, and dominance and orientation maps in the visual cortex[2,74–77]. Together with the metabolic and computational advantages of modular networks, this has raised the possibility that modularity may have particular functional significance for human cognition[22]. For example, it has been suggested that the coding of visual information by multiple cells in a columnar module that have similar but not identical selectivity may arise as a solution to one of the critical problems in object recognition: achieving invariant yet precise representations[28,30,78]. In our data, we find that multiple modules contain category information that is largely independent, and that within modules category decoding is much more similar, although the information is not fully redundant across individual neurons. This is consistent with the proposition that the general class of an object can be represented by the activity of multiple modules whereas precise discriminations may rely on detecting differences in the activity of neurons belonging to the same module.

In closing, our results rest upon converging evidence of both connectivity and function and show that local functional circuits in the human anterior temporal lobe have a modular organization. The existence of functional modules that approximate the size and geometry of cortical columns identified in the human primary visual cortex, suggests that modularity may be a general organizing principle for neural circuits at the micro-scale.

## Methods
### Participants
Eight participants (4 female; 39.3 ± 9.5 years old; mean ± SD) with drug resistant epilepsy underwent a surgical procedure in which platinum recording contacts were implanted on the cortical surface as well as within the brain parenchyma. In each case, the clinical team determined the placement of the contacts to localize epileptogenic regions. In all the participants investigated here, the clinical region of investigation was the temporal lobes.

In each participant, for research purposes we placed one or two microelectrode arrays (MEAs; Cereplex I; Blackrock Microsystems, Inc., Salt Lake City, UT) in the anterior temporal lobe (ATL) in addition to the subdural contacts. When using a single MEA, the MEA is comprised of 96 microelectrode contacts arranged in a 4 × 4 mm grid. For participants with two MEAs, each MEA is comprised of 64 microelectrode contacts arranged in a 3.2 mm × 3.2 mm grid and both MEAs are placed within 1–2 cm of one another. The length of each microelectrode in the MEAs is 1 mm or 1.5 mm. Given an average cortical thickness of around 3 mm, we estimate that the microelectrode tips likely lie in the vicinity of layers 3 or 4. We implanted MEAs only in participants with a presurgical evaluation indicating clear seizure localization in the temporal lobe and the implant site in the ATL was chosen to fall within the expected resection area. Each MEA was placed in an area of cortex that appeared normal both on the pre-operative MRI and on visual inspection. Across participants, MEAs were implanted 19.1 ± 13.6 mm away from the closest subdural electrode with any ictal or interictal activity identified by the clinical team. Five out of the eight participants received a surgical resection which includes the tissue where the MEAs were implanted. None of the participants had intracranial EEG recordings that suggested that the source of seizure activity was in the anterior lateral temporal cortex where the arrays were implanted, yet these regions were removed as part of the standard surgical procedure. While we cannot be certain that these regions are not pathologic in some way, the clinical recordings suggest that they are not directly involved in the seizures that were captured. One participant had evidence of focal cortical seizure activity and received a localized resection posterior to the MEA site. Two participants did not have a sufficient number of seizures during the monitoring period to justify a subsequent resection. Neither participant experienced a change in seizure type or frequency following the procedure, or experienced any noted change in cognitive function.

Data were collected at the Clinical Center at the National Institutes of Health (NIH; Bethesda, MD). The Institutional Review Board (IRB) approved the research protocol, and informed consent was obtained from the participants and their guardians.

### Microelectrode recordings and preprocessing
**Local field potentials.** We digitally recorded microelectrode signals at 30 kHz using the Cereplex I and a Cerebus acquisition system (Blackrock Microsystems, Salt Lake City, UT), with 16-bit precision and a range of ±8 mV. To obtain local field potentials (LFPs), we low-pass filtered the signals at 500 Hz (anti-alias), downsampled to 1000 Hz, and then re-referenced each time series by subtracting the average signal across all microelectrode channels. We used the Chronux 2.11 toolbox to apply a local detrending procedure to remove slow fluctuations (≲2 Hz) from each electrode's time series and used a regression-based approach to remove line noise at 60 Hz and all harmonics up to the Nyquist frequency[79]. During this step we first upsample the signals to 1020 Hz so that the sampling rate is an integer multiple of 60 and then downsample back to the original sampling rate after the regression. These techniques do not suffer from some of the distortions induced by high pass or notch filtering field potential signals[79]. We minimally process the data and avoid unnecessary filtering since excessive filtering and pre-processing can have a deleterious effect on estimating the pair-wise conditional Granger causalities[49]. Supplementary Fig. 1b shows the effect of pre-processing on the power spectrum for an example channel and session.

We collected microelectrode recording data from experimental sessions during which the participants performed a variety of

cognitive testing tasks designed to test their memory or their ability to correctly categorize visual stimuli. We analyzed data from these experimental sessions since this ensures that the participants were awake and actively engaged throughout the duration of each session. We restricted our analysis to sessions with a duration of at least 30 min. Since some sessions lasted longer than 30 min, we only analyzed the first 30 min of each session such that the time-scales for analysis are the same for all participants. We discarded any sessions that contained clear artifacts throughout. We also discarded all experimental sessions that involved direct electrical stimulation so as to avoid electrical artifacts. In total, we captured data from $3.6 \pm 1.2$ sessions per participant, and each experimental session was separated from the next session by an average of $2.1 \pm 2$ days.

For each session, we rejected electrodes if they exhibited abnormal amplitude, variance, or large line noise. In every individual session, we calculated the root mean square amplitude and the variance of the voltage values for each electrode. Any electrode with a voltage trace whose RMS amplitude or variance was greater than three standard deviations away from the mean across all electrodes was flagged for visual inspection. We rejected any electrode which displayed clear artifacts throughout the session and excluded that electrode from the global average when re-referencing[80]. We also inspected time-frequency spectrograms for each electrode after removing line noise, and manually rejected any electrode still exhibiting significant line noise power. We discarded an electrode from all sessions if we rejected that electrode in any one session. After this procedure, we retained $87.5 \pm 0.7$ electrodes from the 4 mm × 4 mm arrays for subsequent analysis, and $51.5 \pm 4.2$ electrodes from the 3.2 mm × 3.2 mm arrays.

Large non-stationary deflections in the time series, such as artifacts or epileptiform discharges, are problematic for computing Granger causality since some of the assumptions of vector autoregressive models are violated[46]. To further mitigate any confounds due to artifacts and epileptiform discharges, we split each session into twenty 30-second clips and noted any time points where the voltage exceeded four scaled median absolute deviations[81]. For each of these time points we removed the adjacent five samples on each side and interpolated using a piecewise cubic Hermite interpolating polynomial[82,83]. From each 30-second clip we extracted the 5-second window with the fewest number of time points that had to be interpolated. Therefore, we retained twenty 5-second blocks in each experimental session that we used for connectivity analysis, with an average inter-block interval of $77.3 \pm 0.4$ s across sessions. See Supplementary Fig. 1d for an example of a single block of processed LFP data sorted by module membership. For the purposes of our analyses, we performed the pseudo-random selection of the individual blocks from the first 30 min of each session without considering any time-locking to any external stimulus, event, or task. Thus, the data in the blocks used for our analyses reflect spontaneous, though not necessarily resting, activity.

**Single unit activity.** For five of the participants we manually identified single units offline that were recorded during a visual categorization task. We used quantitative metrics of isolation quality to estimate the quality of each unit used in subsequent analysis. After identifying a list of channels with potential single-unit activity, we loaded the globally re-referenced and bandpassed (0.3 to 3 kHz) time series of each channel, one at a time, into Plexon Offline Sorter (Plexon, Inc., Dallas, TX) for manual spike sorting. We converted the continuous-voltage time series into a population of voltage snippets (1.067 ms long, 30 samples) that crossed a manually defined voltage threshold. We set the threshold such that random noise fluctuations in the signal would occasionally cross the threshold and be captured as a noise snippet. We projected each snippet into principle component space and only retained isolated units that were separable from each other and from

noise throughout the duration of the experiment[84]. We identified a total of 1034 unique units across 10 recording sessions ($103 \pm 47$ units per recording session) with an average spike rate of $1.47 \pm 0.66$ sp/s and an isolation score of $0.89 \pm 0.06$[85] (Supplementary Fig. 9). We henceforth refer to these isolated units as spiking neurons.

## Directed functional connectivity

We constructed an adjacency matrix for each MEA in each experimental session by treating each microelectrode as a node and computing the pair-wise conditional Granger causality between every pair of microelectrodes (see Supplementary Text)[46,49,86]. The physical nature of structural connections in the brain makes neural networks inherently directed and introduces temporal delays when information is being transferred through these connections. Thus, given recordings of sufficiently high temporal resolution, efforts should be made to capture this feature of brain networks in measures of functional connectivity. Substantial work has demonstrated that Granger causal methods are adequate for detecting these relationships between cortical regions[69], reduce the effect of instantaneous field effects[47], and show higher correspondence to anatomical connectivity compared to other measures of functional connectivity[48].

The Granger causality from $Y$ to $X$ quantifies the degree to which the activity in $Y$ can predict the activity in $X$ beyond the degree to which the activity in $X$ is predicted by its own past. These notions of prediction and precedence in Granger causal analysis can be quantified using vector autoregressive (VAR) modeling. In order to eliminate spurious causalities that may arise due to common dependencies on a third variable, $Z$, we compute the conditional Granger causality which is defined as the conditional $\mathcal{F}$-statistic:

$$\mathcal{F}_{y \to x|z} \doteq \ln \frac{\text{var}(\varepsilon'_{x,t})}{\text{var}(\varepsilon_{x,t})} \tag{1}$$

where $\varepsilon_{x,t}$ is the residual of the full vector autoregressive model that predicts the activity of electrode $X$ based on the past of electrodes $X$, $Y$, and $Z$, and where $\varepsilon'_{x,t}$ is the residual of the reduced model which only uses the past of $X$ and $Z$. If the variability of the residuals in the full model is significantly less than that of the reduced model then the inclusion of $Y$ improves the prediction of $X$.

In a multivariate setting one can condition not just on a single variable $Z$, but on all other known variables besides $X$ and $Y$. This defines the pair-wise conditional Granger causality, $\mathcal{F}_{y \to x|[xy]}$ where $[xy]$ denotes conditioning on all variables besides $x$ and $y$. For simplicity, we use $\mathcal{F}$ or $\mathcal{F}_{y \to x}$ to denote the pair-wise conditional Granger causality. Pair-wise conditional Granger causality quantifies the degree to which activity of a source electrode can predict the activity of a second target electrode over and above the degree to which the target electrode's activity is already predicted by its own past and the past of all other electrodes besides the source[46]. This helps mitigate any spurious causalities that may arise due to common dependencies or volume conduction.

We used the values $\mathcal{F}_{j \to i}$ as connection weights to construct an adjacency matrix for each MEA in each experimental session. Given each $\mathcal{F}$-statistic, we determined whether each connection weight is significantly different from zero and applied a Bonferroni correction to each of the adjacency matrices[49]. We set entries in the adjacency matrix whose connection weights are not significant to zero, and kept the raw values for all entries deemed significant. Thus, for each array and session we constructed a weighted and directed network representation (Fig. 1c).

**Spectral decomposition of conditional Granger causalities.** The framework described above can be implemented in the frequency domain and conditional Granger causalities can be calculated as a function of frequency for each electrode pair[46,87,88]. The fundamental

spectral decomposition in the conditional case is given by:

$$\frac{1}{2\pi} \int_0^{2\pi} f_{y \to x|z}(\omega)\, d\omega = \mathcal{F}_{y \to x|z} \tag{2}$$

In this way, band-limited information can be obtained by making use of:

$$\frac{1}{B} \int_B f_{y \to x|z}(\omega)\, d\omega = \mathcal{F}_{y \to x|z}(B) \tag{3}$$

where $B$ is the band of interest. Because modularity is maximized by grouping together electrodes with stronger functional connections, the spectrum Granger spectrum for electrode pairs within the same module could be biased to exhibit an overall positive shift compared to the spectrum for for electrode pairs that span different modules. As such, we normalized the individual spectra for each electrode pair to have a total area of one, allowing us to compare the relative frequency contributions in each electrode pair.

## Multilayer modularity optimization

To track how the modular structure of local networks evolves over time we adopted a multilayer modularity approach. First, we stacked the adjacency matrices from all sessions for a single participant in chronological order. This can be represented by a rank-3 adjacency tensor, $[A]_{i,j,s}$, where $i, j$ are the indices for the electrodes and $s$ denotes the individual sessions (or layers). Hence, $A_s$ represents the $N \times N$ adjacency matrix for session $s$ where $N$ is the number of electrodes in an array. This network representation is also known as a temporal network[51,52]. Next, for every electrode in layer $s$ we introduced links of weight $\omega$ from that electrode to itself in layer $s + 1$ (black arrows in Fig. 1e). This type of coupling has been referred to as diagonal (across-layer edges only from an electrode to itself), ordinal (only edges between consecutive layers), and uniform (same edge weight, $\omega$, for all across-layer edges) interlayer coupling[54,89,90]. For each array we set $\omega$ to be equal to the median non-zero weight across all sessions so that the across-layer weights are of the same order as the within-layer weights (Supplementary Fig. 6).

Once the functional networks have been cast into a multilayer representation we can find the optimal partitioning across all sessions by optimizing the multilayer quality function $Q_{ml}$[53]:

$$Q_{ml} = \frac{1}{\mu} \sum_{ijst} [(A_{ijs} - \gamma P_{ijs})\delta_{st} + \delta_{ij}\omega_{jst}]\delta(c_{is}, c_{jt}) \tag{4}$$

were $\mu$ is the total summed weight across all edges and $P_{ijs}$ is the expected connection strength for electrodes $i$ and $j$ in layer $s$ under the Leicht-Newman null hypothesis for directed networks[91]. $c_{is}$ denotes that electrode $i$ belongs to community $c$ in layer $s$, and $\delta_{ij}$ is the Kronecker delta. $\gamma$ is the resolution parameter, which we set to the default value of 1 (Supplementary Fig. 6). Intuitively, the quality of the optimal partition, $Q_{ml}^{max}$, also known as the modularity of the network, tells us the fraction of the total edge weight that falls within groups minus the expected fraction if the edges were distributed randomly[3]. Modularity has an upper bound of 1, and in practice, values larger than 0.3 are taken as evidence for a modular network[55]. A major advantage of this approach is that a module that exists in two different layers will be consistently labeled, avoiding the need to match communities that would arise if modularity was estimated for each session separately.

Because modularity optimization has been shown to be computationally intractable for most networks ($\mathcal{NP}$-hard)[92], computational heuristics must be applied to find approximate solutions. To this end, we used the generalized Louvain algorithms[93]. The quality of the optimal partition is nearly-degenerate in that many partitions exist whose quality value is very close to that of the true optimum. The number of such partitions is bounded by $2^{\bar{n}-1}$ from below and the $\bar{n}$th Bell number from above, where $\bar{n}$ is the average number of modules[94]. For the networks considered here $\bar{n} = 6.1 \pm 0.9$. Because the generalized Louvain algorithm is non-deterministic, we repeated the Louvain algorithm 500 times for each MEA in order to adequately sample the distribution of $Q_{ml}^{max}$ values. We also systematically varied the resolution parameter $\gamma$ and the between-layer coupling parameter $\omega$ to confirm that this did not have a significant effect on our main results (Supplementary Text and Supplementary Fig. 6).

To estimate the spatial extent of each module, we computed the root mean squared distance from all nodes within a module to the module's center. This represents the radius of gyration, $R_g$, for a set of equal masses. We used the module diameter, $2R_g$, as the measure of the module's spatial extent. To quantify the spatial contiguity of each module, we first calculated the convex hull for each module, and then defined the spatial compactness of that module as the proportion of nodes within the hull that belong to the module out all the nodes contained within the hull (Fig. 2f). Hence, if the nodes in a module tend to be spatially contiguous, then the average spatial compactness of that module over all sessions should be near one.

## Visual categorization task

Five participants performed an image categorization task that required them to view a series of images and to report whether each image was a person, place, object, or animal using the arrow keys (Supplementary Fig. 7a). We presented the task on a laptop that was synchronized with the data acquisition system used to record MEA data. Each MEA was implanted in the anterior temporal lobe, a region often implicated in processing semantic representations, particularly of famous people and places[63-65]. We therefore used an image set consisting of 60 images each of famous people (e.g., Brad Pitt, Barak Obama), famous places (e.g., Eiffel Tower, Niagara Falls), animals (e.g., raccoon, flamingo), and man-made objects (e.g., light bulb, hammer). We obtained images from free image searches online and the images had a range of different background detail, ranging from full natural scene backgrounds (e.g., a dolphin jumping in the ocean) to portraits on a uniform background. We cropped and resized each image to $1000 \times 1000$ pixels and converted it to grayscale. We then randomly phase-shifted in the frequency domain approximately 90% of the pixels to gently blur each images and soften any individual features that could be used for categorization (e.g., the facial features)[95]. We balanced the image set for luminance, contrast, and spatial frequency using the SHINE toolbox[96]. Each image covered $\approx 50\%$ of the laptop's screen, centered over a gray background.

In each experimental session, we presented the full set of 240 images in pseudorandom order. We presented each image for 500 ms and displayed the four category options in text on the four edges of the screen (top, bottom, left, and right) that corresponded to that option's arrow key. Once the participant made a selection on each trial, the screen went blank for 200 ms before the next image appeared. Every 60 images the task automatically paused and reported the score to the participant (e.g., '51 of 60 correct'). We defined the baseline period of the task as the 2000-ms interval preceding and following each set of 60 presented images, during which time we presented an image of fixed pattern noise on the screen in the same location as the categorical images. Each participant completed $2 \pm 0.75$ sessions (240 to 1680 trials per session). Performance was $\geq 88\%$ (range 88 to 97%) and on average median response time was $1053 \pm 270$ milliseconds across all participants (Supplementary Fig. 7b, c).

## Image classification using single unit activity

We considered a neuron to be a member of a module if it was recorded from a microelectrode that was assigned to that module based on LFP signals. The LFPs used to define the modules were typically extracted from different sessions than the single unit activity used for image

classification (10 categorization task sessions used for single unit analysis, 20 sessions used for modularity analysis, and 4 sessions used for both), hence we assign neurons to modules based on the closest session in time. We only analyzed neurons that were recorded on a microelectrode that was considered part of a module, and we only analyzed modules for which we captured spiking activity from two or more neurons. Across MEAs and behavioral sessions, we recorded spiking activity from neurons belonging to 75 modules (median 8 neurons per module range 2 to 34; median 4.25 modules per MEA, range 0 to 7; see Supplementary Fig. 8).

For visualization, we created a population spike raster for each trial by aligning the spike times from each recorded neuron to the image onset for that trial, and then separating the raster into different modules. For classification analysis, we transformed spike times into z-scored instantaneous spike rates using a 200-ms boxcar sliding window, with steps of 20 ms. Due to the low spike rates of human temporal lobe neurons, we square-root transformed spike counts in each 200-ms bin. We then converted the spike rates to a z-transformed, baseline-corrected spike rate by subtracting the mean and dividing by the standard deviation of the square root spike counts during the baseline period[66].

We used logistic regression classification with early stopping to predict image category from population spiking activity[84,97]. We only used trials in which the participant correctly identified the image category. We used twenty-fold cross validation to estimate prediction accuracy. Each hold-out set was made up of approximately 5% of all trials uniformly distributed across the session. For each fold, one quarter of the training trials were randomly selected to be used as the early stopping test set. To avoid overfitting, we iteratively evaluated this test set to determine when to stop annealing the regression weights. We used the remaining training trials to z-score the training, early stopping, and hold-out data. We repeated the random selection of early stopping trials 50 times per fold, and computed the average resultant weights for each fold before computing prediction accuracy on the hold-out set. We separately performed this procedure using the instantaneous rate of the neurons in each individual time bin from −0.2 s to 1.2 s relative to image onset in order to visualize the time-course of image category discriminability. Using this approach, the number of predictors used for classification at each time point is therefore equal to the number of neurons. Using the same procedure, we also built a single classifier that estimated overall classification accuracy for the entire response period using the aggregated spiking data from every third time bin from 0.1 s to 1.2 s. In this case, the total number of predictors used for classification is therefore the number of neurons multiplied by the 18 selected time bins. We tested for significant classification by comparing overall classification accuracy to the overall accuracy of a matched surrogate. In this case, for each surrogate, we calculated classification accuracy using exactly the same procedure except that we randomly shuffled the trial-labels of the four categories for the training and early-stopping trials. We considered overall classification significant if a paired t-test between the 20 true and trial-shuffled accuracy measurements were significantly different at $p \le 0.05$ (two-tailed test).

We computed classifier weights and z-score parameters separately for each module to determine whether the neurons in that module significantly coded image categories. We used those same classifier parameters to quantify the relative similarity of neural coding among neurons within the same module versus neurons from different modules by strategically shuffling the hold-out data used to estimate classification accuracy. When shuffling within modules, we randomly permuted the label of the neurons in the hold-out data so the classifier weights were applied to data from a different neuron from the same module when computing classification accuracy for each fold. We repeated this shuffling procedure 100 times, averaging the accuracy within folds and then testing for

significance across folds versus a matched surrogate. When shuffling across modules, we followed a similar procedure, except in this case there was a chance of mismatch between the number of units in the module used to build the classifier versus the number of units in the module contributing data to the hold-out set. If there were fewer units in the classifier module, we randomly selected a sub-sample of units for the hold out set during each of the 100 iterations. Similarly, if there were more units in the classifier module, we randomly resampled neurons from the hold out set. We tested for significant differences between shuffle-within and shuffle-across classification accuracy using a paired t-test across modules, where each module contributed a shuffle-within score and a shuffle-across score. We also computed the average shuffle-within and shuffle-across accuracy for all modules recorded from an array on a given behavioral session, and computed a paired t-test on the distribution of accuracies across arrays and sessions between the two conditions.

## Statistics
Results are reported as mean ± SD unless otherwise noted. We provide statistics for modularity, stationarity and persistence for all arrays as well as across all arrays in Supplementary Table 1. Similarly, we provide statistics for module diameter and compactness in Supplementary Table 2. We used a significance threshold of $p = 0.05$ for all statistical tests.

In order to compare the functional connections and modules observed in each MEA to the connections and modules expected by chance, we used a shuffling procedure. In brief, for each MEA in each experimental session, we generated 500 surrogate networks. We constructed each surrogate network by randomly swapping links between the microelectrodes while ensuring that the distribution of degrees matched the distribution of in and out degrees present in the true networks. We performed this random shuffling 100 times for each surrogate, and used the final iteration as the random network for that surrogate. We compared the pairwise similarity between the true MEA networks across all experimental sessions to the pairwise similarity between the random surrogate networks for that MEA. Similarly, we compared the quality of the optimal partition (modularity), the stationarity, the persistence, and the average module diameter and compactness in the true data for each MEA to each of these measures for the surrogate networks.

### Reporting summary
Further information on research design is available in the Nature Research Reporting Summary linked to this article.

## Data availability
The data that support the findings of this study are available at https://research.ninds.nih.gov/zaghloul-lab/downloads and also from the corresponding author upon reasonable request. Source data are provided with this paper.

## Code availability
Except where otherwise noted, computational analyses were performed using custom written MatLab (MathWorks) scripts. The custom MATLAB scripts used for analysis are available upon request.

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

## Acknowledgements
We thank Vishnu Sreekumar and Weizhen Xie for helpful and insightful comments on the manuscript. This work was supported by the Intramural Research Program of the National Institute for Neurological Disorders and Stroke, and utilized the computational resources of the NIH HPC Biowulf cluster (http://hpc.nih.gov). We are indebted to all patients who have selflessly volunteered their time to participate in this study.

## Author contributions
Conceptualization, J.I.C. and K.A.Z.; Methodology and Software, J.I.C. and J.H.W; Formal Analysis, J.I.C. and J.H.W.; Investigation, J.I.C., J.H.W., S.K.I., K.A.Z.; Writing, J.I.C., K.A.Z., and J.H.W.; Visualization, J.I.C. and J.H.W.; Supervision, K.A.Z.

## Funding

## Competing interests
The authors declare no competing interests.
