## [Peer Review File · Nature Communications]

Micro-scale functional modules in the human temporal lobeREVIEWER COMMENTS

Reviewer #1 (Remarks to the Author):

In this report, Chapeton et al. aimed to address a fascinating open question in the field of system neuroscience, namely if there is a modular organization in higher association cortices akin to the one observed in primary sensory cortices. They measured a combination of local field potentials (LFPs) and neuronal firing rates using microelectrode arrays (MEAs) implanted in the middle temporal gyrus. They partitioned the directed functional networks using the LFP signals and then showed that functional modules indeed exist and exhibit independent functional properties based on the spiking responses of individual neurons.

I find their findings extremely important and novel and in line with several lines of evidence from other intracranial studies, including our own. Their novel findings are unique and essential for the field of human intracranial electrophysiology.

The finding of strength and direction of the identified functional connections staying stable across several days, and membership in each module not changing substantially from one time to the next is supported by what has been historically reported with iEEG and single-unit in non-human and human recordings. From day to day, the responses of a patch of cortex remains stable.

Their finding that shuffling neuron identity within each module decreased classification accuracy, but the classification remained more significant than chance is also very interesting since it explains why we get the same response by inserting the electrode in the same area of the cortex even though we certainly are not hitting the same neurons each time. This finding is in line with Yuste's well known critical view of single unit theories of cognition (Yuste Nature Reviews Neuroscience 2015).

While I think the present work is elegantly presented, I admit that I am not a network statistician, and I do not do Granger kind of analyses and as such the rigor of the statistical analysis ought to be vetted by an expert reviewer.

I like to comment that the discussion is shallow and too short and redundant with the introduction. I wished to see a more in-depth discussion of their observations' methodological limitation (for the most part, one single array was implanted in the cortex, and there is no information about how these so-called modules are organized in a patch of larger cortex (i.e., in a supra-millimeter space). For this, authors could have linked their findings to those by Eddie Chang or Bob Knight's earlier studies showing how a patch of cortex responds across a high-resolution grid of macro electrodes (within 2mm you get

different response - see for example Adeen Flinker's work). Also, ideas for future research and implications of their findings for neuroscience ought to be discussed.

Minor Suggestions:

Line 22 and 223: Reference 38 should be changed to

1. Schrouff et al. Fast temporal dynamics and causal relevance of face processing in the human temporal cortex. *Nature communications*. 2020 Jan 31;11(1):656.
2. Parvizi J, Jacques C, Foster BL, et al. Electrical stimulation of human fusiform face-selective regions distorts face perception. *J Neurosci*. 2012 Oct 24;32(43):14915-20.

Authors provided information about the size and location of a specific patch of FFA in Schrouff et al. The current reference to Rangarajan is not a correct one.

Line 22-23 (and also 217-228): Please include a reference and mention Dastjerdi M, Ozker M, Foster BL, et al. Numerical processing in the human parietal cortex during experimental and natural conditions. *Nature communications*. 2013. In this study, the authors show how a distinct population of neurons under a specific electrode has a selective arithmetic function during experimental and real-life unconstrained settings from one day to another and shows how a 2x2mm area of cortex exhibits very unique function across task and real life testings from day to day + its adjacent sites are not responding the same way at all.

Also additional supporting evidence comes from electrical stimulation of the brain. For instance, stimulations have different effects when you move from one site to another that is only 2-3mm apart (edge of electrode to edge of adjacent electrode). This has been reported in almost every stimulation study by Parvizi lab.

Reviewer #2 (Remarks to the Author):

The authors utilized microelectrode array recordings in the human anterior lobe cortex to examine whether electrodes clustered into spatially contiguous functional modules. They found that, based on local field potential (LFPs) activity recorded during various cognitive tasks, electrodes could be clustered into modules of ~1.3mm diameter size. They further found that neurons within these LFP-defined modules had more similar tuning to each other, and that boundaries between modules were sharp.

This work is very interesting, novel, done in a rigorous way, and of high significance. It presents to my knowledge the first direct evidence for anatomically defined functional modules in areas outside of primary sensory areas in humans. The work is based on the very rare ability of multiarray recordings in humans with fixed pitch, allowing continuous sampling of defined pieces of cortex (here, 4x4mm) at microscale resolution. These recordings are performed in anterior temporal lobe, an area of significant interest for semantic memory. Overall, this paper presents very rare and precious new data combined with a rigorous analysis providing significant new insight into the organization of the human brain. I raise below a few issues that I had questions about when reading the manuscript. I recommend publication in this journal after addressing these.

Major issues:

1. Definition of modules based on LFP. What aspect of the LFP define the modules? If I understand correctly, the GC-based connectivity matrices were defined based on broadband activity (what was the high-pass filter?). First, it would be helpful to visualize the characteristics of these signals after all pre-processing (spectra) so that one can see what signals defined the modules. Also, could one appreciate modules when looking at the raw traces sorted by module identity? Second, since the signal is $1/f$ distributed, presumably the lower frequencies were what defined the modules. Is this true and could the same or different modules be defined when using activity in defined frequency bands (theta, beta, gamma etc) ? This is the most significant uncertainty I have re the paper - what defines the modules and how sensitive is this to specific choice of LFP features used.

2. The authors examined the tuning of individual neurons with respect to modules independently defined based on LFP data. The principle support for their argument that the tuning of spiking responses with respect to categories clusters within these modules is that decodability decreases to chance when neurons are scrambled across clusters, but not within clusters (Fig. 3e). But this leaves open the question of why decoding did not generalize to outside the module. Is it because the proportion of neurons that is tuned to category differs between modules or because they are tuned in similar proportions but to different categories? A single-neuron analysis of category selectivity as routinely done for MTL (select cells with ANOVA, assess preferred category, depth of selectivity) should shed light on this (i.e. Minxha et al. 2017, Cell Reports).

3. Was the definition of what constitutes a module sensitive to whether this analysis was performed during a period after stimulus onset (such as the one used to define category selectivity in Fig. 3) or could modules also be defined based on spontaneous activity (during baseline)?

4. What was the effect of the limited sampling (i.e. 4x4mm patch) on the estimated module size? presumably the size is under-estimated due to this edge effect? A simple simulation could answer this question (or comparison with the 3.2mm² vs 4mm² data?).

Minor issues:

1. Can anything be said from what layer the recordings likely took place?

Reviewer #3 (Remarks to the Author):

[see attachment]

Manuscript#: NCOMMS-21-43959

Title: Micro-scale functional modules in the human temporal lobe

The manuscript presents results from functional connectivity (FC) and graph theoretical analyses of neurophysiological data measured in the human temporal cortex of patients using intracranial microelectrode recordings. The results highlight a spatial modular organisation in temporal graphs constructed from FC metrics. These modules are temporally persistent and spatially compact, approximately 1.3mm in diameter. In addition, neurons within modules share more information regarding the semantic category of the image than neurons from different modules. These results are interpreted as suggesting the presence of anatomo-functional modules at the microscale in the human temporal cortex. Although I find that the computational tools used to analyse the neural data are appropriate, I have major concerns about the framing of current knowledge about the neural bases of visual processing (visual categorisation) and the interpretation of the results.

1. The manuscript presents original analyses of MEA recordings, but it is highly detached from results in the literature regarding the role of temporal cortex in object recognition. For example, the manuscript completely lacks a state-of-the-art regarding the current knowledge in the field including potential models of functional columnar organisation. Some examples are:

- Tanaka, K. Inferotemporal cortex and object vision. *Annu. Rev. Neurosci.* 19, 109–139 (1996).
- Logothetis, N. K. & Sheinberg, D. L. Visual object recognition. *Annu. Rev. Neurosci.* 19, 577–621 (1996).
- DiCarlo, J. J., Zoccolan, D. & Rust, N. C. How does the brain solve visual object recognition? *Neuron* 73, 415–434 (2012).

Here are some references regarding the potential columnar functional organisation in the temporal cortex:

- <https://academic.oup.com/cercor/article/19/8/1870/412198>
- <https://www.nature.com/articles/s41467-020-17714-3>

My first major concern is the lack of reference to current literature about the functional organisation of the temporal cortex . The manuscript states in the introduction that the aim is to “test the hypothesis that the human temporal association cortex, a higher-order brain region involved in semantic processing, exhibits a modular organisation at the micro scale”. However, no hypotheses are taken from current literature.

2. The modularity analysis is performed on directed temporal networks computed from Granger causality measures, which quantify information flow rather. The rationale for such analysis is not clearly stated. Why did you privilege the analysis of directed rather than undirected FC matrices, as computed for example by means of spectral coherence? Did you try to reproduce the results using different FC metrics?

3. The Louvain algorithm for modularity maximisation is a nondeterministic heuristic, so it needs to be initialised with random seeds. In addition, it depends on the resolution parameter gamma, which controls over the size and number of communities found (resolution equal to 1 leads to the standard Louvain method, whereas higher and lower resolutions produce larger and smaller number of clusters, respectively). It is not clear why the gamma parameter was set to 1. This is a crucial point of the study, because the major claim concerning the presence of modules of size 1.3mm strongly depends on such parameters. How can you exclude that your main claim about the spatial resolution of functional modules is not a by-product of the parameters of the analysis? What would have happened for example if the size of the MEA would have been 40mm in size rather than 4mm? We may argue that the Louvain method would have found approximately 5-7 clusters, thus giving a spatial resolution approximately 13mm (ten times as the current one). To conclude, my major concern is the indirect measure of functional modules from co-variance based temporal graphs, rather than direct neural measures.

4. As far as I understand, the manuscript presents results from graph theoretical analysis of Granger causality estimates computed over neural signals both during the baseline and task-related period of the visual categorisation task. If this is the case, the Granger causality estimates reflect both task-related and baseline activity. Can you explain the rationale for such a choice? Normally, task-related Granger causality displays large modulations with respect to baseline. Such differences may lead to differences in modularity analysis. Can you still claim that the modules are exclusively related to task-related activity.

5. Granger causality analysis is performed on broad-band LFPs. Current theories about brain communication through coherence, however, provide diverging hypotheses about the role of different frequency bands (alpha/beta versus gamma) in brain communication. One of the most recent studies in the field (Vezoli et al Neuron 2021, <https://www.sciencedirect.com/science/article/pii/S089662732100725X>) suggests that different frequency bands have different roles and different modular organisation. Can you

please explain the rationale for taking broad-band LFP for the calculation of Granger causality?

6. Five out of the eight participants received a surgical resection which includes the tissue where the MEAs were implanted. Can you please comment on the fact that most of the recorded neural tissue was pathological and needed resection?

I hope the points I raised will help ameliorating the quality of the study.

Best regards.

Reviewer #1 (Remarks to the Author)

In this report, Chapeton et al. aimed to address a fascinating open question in the field of system neuroscience, namely if there is a modular organization in higher association cortices akin to the one observed in primary sensory cortices. They measured a combination of local field potentials (LFPs) and neuronal firing rates using microelectrode arrays (MEAs) implanted in the middle temporal gyrus. They partitioned the directed functional networks using the LFP signals and then showed that functional modules indeed exist and exhibit independent functional properties based on the spiking responses of individual neurons.

I find their findings extremely important and novel and in line with several lines of evidence from other intracranial studies, including our own. Their novel findings are unique and essential for the field of human intracranial electrophysiology.

The finding of strength and direction of the identified functional connections staying stable across several days, and membership in each module not changing substantially from one time to the next is supported by what has been historically reported with iEEG and single-unit in non-human and human recordings. From day to day, the responses of a patch of cortex remains stable.

Their finding that shuffling neuron identity within each module decreased classification accuracy, but the classification remained more significant than chance is also very interesting since it explains why we get the same response by inserting the electrode in the same area of the cortex even though we certainly are not hitting the same neurons each time. This finding is in line with Yuste's well known critical view of single unit theories of cognition (Yuste Nature Reviews Neuroscience 2015).

While I think the present work is elegantly presented, I admit that I am not a network statistician, and I do not do Granger kind of analyses and as such the rigor of the statistical analysis ought to be vetted by an expert reviewer.

We thank the Reviewer for these encouraging and overall positive comments.

I like to comment that the discussion is shallow and too short and redundant with the introduction. I wished to see a more in-depth discussion of their observations' methodological limitation (for the most part, one single array was implanted in the cortex, and there is no information about how these so-called modules are organized in a patch of larger cortex (i.e., in a supra-millimeter space). For this, authors could have linked their findings to those by Eddie Chang or Bob Knight's earlier studies showing how a patch of cortex responds across a high-resolution grid of macro electrodes (within 2mm you get different response - see for example Adeen Flinker's work). Also, ideas for future research and implications of their findings for neuroscience ought to be discussed.

We agree with the Reviewer that our initial manuscript could benefit from a more expanded discussion of these very relevant points. We have now revised our Discussion accordingly, including relevant references and implications for future work.

Minor Suggestions:

Line 22 and 223: Reference 38 should be changed to

- 1. Schrouff et al. Fast temporal dynamics and causal relevance of face processing in the human temporal cortex. *Nature communications*. 2020 Jan 31;11(1):656.**
- 2. Parvizi J, Jacques C, Foster BL, et al. Electrical stimulation of human fusiform face-selective regions distorts face perception. *J Neurosci*. 2012 Oct 24;32(43):14915-20.**

Authors provided information about the size and location of a specific patch of FFA in Schrouff et al. The current reference to Rangarajan is not a correct one.

Line 22-23 (and also 217-228): Please include a reference and mention Dastjerdi M, Ozker M, Foster BL, et al. Numerical processing in the human parietal cortex during experimental and natural conditions. *Nature communications*. 2013. In this study, the authors show how a distinct population of neurons under a specific electrode has a selective arithmetic function during experimental and real-life unconstrained settings from one day to another and shows how a 2x2mm area of cortex exhibits very unique function across task and real life testings from day to day + its adjacent sites are not responding the same way at all.

Also additional supporting evidence comes from electrical stimulation of the brain. For instance, stimulations have different effects when you move from one site to another that is only 2-3mm apart (edge of electrode to edge of adjacent electrode). This has been reported in almost every stimulation study by Parvizi lab.

We thank the Reviewer for pointing out these errors and omissions. We have now carefully reviewed these references. We have included additional relevant references and have also corrected the errors in references identified by the Reviewer.

Reviewer #2 (Remarks to the Author)

The authors utilized microelectrode array recordings in the human anterior lobe cortex to examine whether electrodes clustered into spatially contiguous functional modules. They found that, based on local field potential (LFPs) activity recorded during various cognitive tasks, electrodes could be clustered into modules of ~1.3mm diameter size. They further found that neurons within these LFP-defined modules had more similar tuning to each other, and that boundaries between modules were sharp.

This work is very interesting, novel, done in a rigorous way, and of high significance. It presents to my knowledge the first direct evidence for anatomically defined functional modules in areas outside of primary sensory areas in humans. The work is based on the very rare ability of multiarray recordings in humans with fixed pitch, allowing continuous sampling of defined pieces of cortex (here, 4x4mm) at microscale resolution. These recordings are performed in anterior temporal lobe, an area of significant interest for semantic memory. Overall, this paper presents very rare and precious new data combined with a rigorous analysis providing significant new insight into the organization of the human brain. I raise below a few issues that I had questions about when reading the manuscript. I recommend publication in this journal after addressing these.

We thank the Reviewer for this assessment of our manuscript and for the positive comments.

Major issues:

1. Definition of modules based on LFP. What aspect of the LFP define the modules? If I understand correctly, the GC-based connectivity matrices were defined based on broadband activity (what was the high-pass filter?). First, it would be helpful to visualize the characteristics of these signals after all pre-processing (spectra) so that one can see what signals defined the modules. Also, could one appreciate modules when looking at the raw traces sorted by module identity? Second, since the signal is $1/f$ distributed, presumably the lower frequencies were what defined the modules. Is this true and could the same or different modules be defined when using activity in defined frequency bands (θ , β , γ etc) ? This is the most significant uncertainty I have re the paper - what defines the modules and how sensitive is this to specific choice of LFP features used.

The Reviewer raises several important points regarding how we define our modules based on the LFP signal. As the Reviewer notes, in our primary analysis we use Granger causality to identify directed effective connections between micro-electrodes, and then identify modules within which connectivity is strong and across which connectivity is weaker. The LFP signals that we use to compute these measures are effectively broadband, although we use a local detrending procedure to remove slow fluctuations in the time series. This approach does not suffer from some of the distortions induced by high pass filtering field potential signals (Mitra and Bokil. Observed Brain Dynamics. Oxford University Press, Inc., 2009.). The effects of detrending the time series data are comparable to applying a high-pass filter at ~ 1 -2Hz. We have now clarified this approach in the revised Methods (“Local field potentials” subsection). We have also included an example power spectral density before and after the detrending procedure that demonstrates the main effect of the detrending procedure in attenuating the very low frequencies in the signal (Supplementary Figure S1b).

The Reviewer offers a good suggestion regarding visualizing the signals following pre-processing. We have now included the processed time series from a single example MEA, ordered both by the original electrode number, and then re-ordered and color coded by module membership, in Supplementary Figure S1c,d. As the Reviewer has suggested, simple visual inspection of the time series is sufficient to appreciate that voltage traces from the same module appear similar.

The Reviewer also raises a good point regarding the frequencies that contribute to the connectivity we measure through Granger causality. The pipeline for calculating the pair-wise conditional Granger causality used in the MVGC toolbox involves first computing the Granger causality as a function of frequency, and then integrating over all frequencies to obtain the overall Granger causality. This spectral Granger function can also be integrated over specific frequency ranges to obtain band-specific contributions to the overall Granger causality. To address the uncertainty and concern raised by the Reviewer, we have now completed new analyses in which we examine Granger causality as a function of frequency. We have included the results of these new analyses in a new main figure, Figure 3, along with a new subsection in results (“Frequency specific contributions to within vs across module connectivity”). We show that in a single example MEA, Granger causality is higher at all frequencies for within-module pairs on average compared to across module pairs. We feel that this overall difference justifies the use of the broadband Granger causality. However, if we normalize the individual Granger spectra to remove this overall shift, we can also analyze the relative contributions from different frequencies to the overall Granger causality. As a result, we find that the average frequency spectrum across sessions for within-module electrode pairs demonstrates a larger contribution

from higher frequencies compared to across-module electrode pairs. To quantify this effect across MEAs we aggregated frequencies into five frequency bands: delta (2-4Hz), theta (4-8Hz), alpha (8-16Hz), beta (16-32Hz), and broadband gamma (70-150). Across all MEAs, within-module functional connectivity is mediated by significantly more high frequency activity and significantly less low frequency activity than across-module connectivity. This asymmetry is consistent with models of cortico-cortical communication which rely on oscillations at different frequencies to synchronize neuronal populations at different spatial scales.

2. The authors examined the tuning of individual neurons with respect to modules independently defined based on LFP data. The principle support for their argument that the tuning of spiking responses with respect to categories clusters within these modules is that decodability decreases to chance when neurons are scrambled across clusters, but not within clusters (Fig. 3e). But this leaves open the question of why decoding did not generalize to outside the module. Is it because the proportion of neurons that is tuned to category differs between modules or because they are tuned in similar proportions but to different categories? A single-neuron analysis of category selectivity as routinely done for MTL (select cells with ANOVA, assess preferred category, depth of selectivity) should shed light on this (i.e. Minxha et al. 2017, Cell Reports).

We thank the Reviewer for raising this question. As the Reviewer notes, the ability to decode the spiking responses within modules decreases when neurons are shuffled across modules. However, the question remains as to whether this decrease is because the proportion of neurons tuned to category differs between modules, or because the neurons are themselves tuned to different categories.

It is likely that both factors contribute to this effect. A little over 1/3 of the modules had significant coding, and hence, there are shuffles between modules with neurons that are tuned to category and modules with neurons that are not tuned to category. In these cases, the decrease in classification is due to the different proportion of coding neurons. The fact that that some groups of neurons code for image category while others, even just a few hundred microns away, do not, supports the functional independence between the modules.

We can restrict the analysis only to comparisons between modules that both had statistically significant category classification and therefore similar numbers of coding units. Across all modules from all MEAs that had multiple modules that significantly code for category we find that the effect holds and there is still a drop in classification accuracy that is significant across all modules from all MEAs and sessions ($p < .001$, $t(93) = 8.54$), indicating that even when two modules are tuned to category, they can encode different features. There were 4 sessions with multiple modules within the same MEA that had significant classification, and the module-averaged shuffled within accuracy was higher than the shuffled across accuracy for 3 of the 4 sessions, although the effect was not statistically significant. We have added these results to the Supplementary text.

While this analysis (restricting scrambles to modules with significant classification) does not distinguish whether the lack of generalization in these cases arises from modules coding different features about the same category or from modules responding to different categories, it does provide supporting evidence for our main claim that the individual modules are functionally distinct. The focus of this study is on mesoscopic organization at the level of modules in the temporal lobe; however, the reviewer does raise important and interesting questions regarding the coding properties and category selectivity of individual single units in this brain region. Indeed, our group is currently finalizing a manuscript that does focus on more classic single-

neuron analyses of category selectivity. There, we thoroughly probe the selectivity to individual categories, the proportions of neurons tuned to one or more categories, and the coding strategies of single neurons in this categorization task, although these analyses are outside the scope of the present study.

3. Was the definition of what constitutes a module sensitive to whether this analysis was performed during a period after stimulus onset (such as the one used to define category selectivity in Fig. 3) or could modules also be defined based on spontaneous activity (during baseline)?

We agree with the Reviewer that this is an important point. We extracted the LFPs used to define the modules from experimental sessions during which the participants performed a variety of cognitive testing tasks. Mostly these were experimental sessions in which participants were not in fact performing the category selection task, but instead were engaged in other cognitive tasks. We analyzed data from these experimental sessions since this ensures that the participants were awake and actively engaged throughout the duration of each session. However, for the purposes of our analyses, we pseudo-randomly selected the individual blocks for our analyses from the first 30 minutes of each session with no time-locking to any external stimulus, event, or task. Thus, the modules are indeed defined from spontaneous, though not necessarily resting, activity. We have now clarified this point in the revised Methods (“Local field potentials” subsection).

4. What was the effect of the limited sampling (i.e. 4x4mm patch) on the estimated module size? presumably the size is under-estimated due to this edge effect? A simple simulation could answer this question (or comparison with the 3.2mm² vs 4mm² data?).

This is also an important point, as one concern would be that the size of the modules that we identify may simply be related to the size of the MEAs and the spatial layout of the microelectrodes we use. To address this, we have now completed a new analysis directly comparing the module diameters and spatial compactness extracted from the data captured using the 4 x 4 mm and the 3.2 x 3.2 mm arrays, as the Reviewer has suggested. We now include this result as a new panel in Supplementary Figure 5 and discuss it in the Supplementary text. In general, there does not appear to be a clear systematic difference between the two array types. The diameter for one of the larger 4 x 4 mm arrays (A2) is about the same, and sometimes smaller, as for the 3.2 x 3.2mm arrays. The modules for the other 4 x 4mm array (A1) do have the largest diameter. However, this appears to be related to the fact that the modules in this array are the among the least compact. The array with the second largest module diameter is in fact one of the smaller 3.2 x 3.2 mm arrays (A3), and this array also has the second smallest compactness. Thus, the differences in module diameter tend to arise from differences in compactness rather than differences in array size.

To address this question further, we also performed a multiresolution analysis of the Louvain algorithm that we use to define modularity. This question was also raised by Reviewer 3. Briefly, the resolution parameter gamma used in the algorithm impacts the size and number of modules found. Higher and lower values of gamma produce a larger and smaller number of modules, respectively. We agree that the module size is a critical point of the study, and so it is important to demonstrate that this is not confounded by the choice of parameters. In a new analysis, we now vary gamma from 0 to 2 in steps of 0.2 in order to explore the effects specifically on module diameter. We include the results of this analysis for an example array in Supplementary Figure S6. As expected, for very small values of gamma the module diameter is the size of the array

since all nodes are placed into a single module. However, the module sharply decreases around $\gamma = 0.3-0.4$, and for values of γ from 0.8-2 the average module size remains in the range of $\sim 1-1.5$ mm. This analysis suggests that there is a characteristic size to the identified modules.

Minor issues:

1. Can anything be said from what layer the recordings likely took place?

This is a good question, but unfortunately one that we cannot fully address. The length of each microelectrode in the MEAs that we use is 1mm. Given an average cortical thickness of around 3mm, we would estimate that our microelectrode tips likely lie in the vicinity of layers 3 or 4. We have examined histology of the resected tissue, but have been unable to definitively identify layer information based on histology alone. We have now clarified this in the revised Methods. In addition, in the revised Discussion, we acknowledge this limitation and note that recordings from multiple cortical layers will be required to fully assess the columnar organization of these circuits.

Reviewer #3 (Remarks to the Author)

The manuscript presents results from functional connectivity (FC) and graph theoretical analyses of neurophysiological data measured in the human temporal cortex of patients using intracranial microelectrode recordings. The results highlight a spatial modular organisation in temporal graphs constructed from FC metrics. These modules are temporally persistent and spatially compact, approximately 1.3mm in diameter. In addition, neurons within modules share more information regarding the semantic category of the image than neurons from different modules. These results are interpreted as suggesting the presence of anatomo-functional modules at the microscale in the human temporal cortex. Although I find that the computational tools used to analyse the neural data are appropriate, I have major concerns about the framing of current knowledge about the neural bases of visual processing (visual categorisation) and the interpretation of the results.

We appreciate the Reviewers comments and concerns. We have now introduced new analyses and revised our manuscript to address these concerns.

1. The manuscript presents original analyses of MEA recordings, but it is highly detached from results in the literature regarding the role of temporal cortex in object recognition. For example, the manuscript completely lacks a state-of-the-art regarding the current knowledge in the field including potential models of functional columnar organisation. Some examples are:

- **Tanaka, K. Inferotemporal cortex and object vision. *Annu. Rev. Neurosci.* 19, 109–139 (1996).**

- *Logothetis, N. K. & Sheinberg, D. L. Visual object recognition. Annu. Rev. Neurosci. 19, 577–621 (1996).*

- *DiCarlo, J. J., Zoccolan, D. & Rust, N. C. How does the brain solve visual object recognition? Neuron 73, 415–434 (2012).*

Here are some references regarding the potential columnar functional organisation in the temporal cortex:

- <https://academic.oup.com/cercor/article/19/8/1870/412198> • <https://www.nature.com/articles/s41467-020-17714-3>

My first major concern is the lack of reference to current literature about the functional organisation of the temporal cortex . The manuscript states in the introduction that the aim is to “test the hypothesis that the human temporal association cortex, a higher-order brain region involved in semantic processing, exhibits a modular organisation at the micro scale”. However, no hypotheses are taken from current literature.

We thank the Reviewer for raising these important points. We agree with the Reviewer that it would be critically important to frame our study and results within the current literature. As such, we have now completely revised our Introduction and Discussion. We have carefully reviewed the references suggested by the Reviewer as well as other related references and have incorporated these references and the current knowledge into our revised manuscript. We agree that there has been substantial evidence supporting functional columnar organization. Our interpretation of this work has been that much of this evidence has been derived from animal studies and has focused on primary sensory regions of the cortex. From this large body of work, we focused on the rodent barrel columns since these are the canonical cortical modules and the ocular dominance columns because they are the best described example in humans. However, we agree with the reviewer that there have been systematic studies in monkeys of modular organization in the inferior temporal cortex within the context of visual object recognition, and we have incorporated these studies into our introduction and discussion in order to better frame and motivate our investigation of functional modules in the human temporal lobe.

Nevertheless, the specific functional role of the anterior temporal lobe remains unclear. Several studies have suggested that the anterior temporal lobe may be involved in semantic representations and in memory formation. As such, we have considered this brain region as one involved in higher order cognitive functions. It is certainly possible that functional columnar organization is a universal organizing principle for all brain regions, including those that are involved in higher order cognition, but this has not been explicitly demonstrated in humans. In many ways, we view our results as building upon a formidable body of work supporting functional columnar organization, and our intention here is to provide evidence that this organizing principle may extend to this higher order brain region, the anterior temporal lobe, in humans.

2. The modularity analysis is performed on directed temporal networks computed from Granger causality measures, which quantify information flow rather. The rationale for such analysis is not clearly stated. Why did you privilege the analysis of directed rather than undirected FC matrices, as computed for example by means of spectral coherence? Did you try to reproduce the results using different FC metrics?

The Reviewer raises an important point regarding the choice of Granger causality. There are two primary reasons we elected to use this approach.

First, there is mounting evidence that structural connection patterns are major determinants of the functional dynamics of cortical circuits as captured by measures of functional or effective connectivity. This correspondence is particularly strong when functional networks are constructed or averaged over longer time scales (Honey et al 2007 *Network structure of cerebral cortex shapes functional connectivity on multiple time scales*, PNAS; Sporns and Tononi 2007 *Structural Determinants of Functional Brain Dynamics*; Bullmore and Sporns *Complex brain networks: graph theoretical analysis of structural and functional systems*, Nat. Rev. Neuro; Honey et al 2009 *Predicting human resting-state functional connectivity from structural connectivity*, PNAS; Kramer et al 2011 *Emergence of persistent networks in long-term intracranial EEG recordings*, J. Neuro). The physical nature of structural connections in the brain makes neural networks inherently directed and introduces temporal delays when information is being transferred through these connections. Thus, given recordings of sufficiently high temporal resolution, efforts should be made to capture this feature of brain networks in measures of functional connectivity. Substantial work has demonstrated that Granger causal methods are adequate for detecting these relationships between cortical regions (reviewed in Bastos, Vezoli, Fries 2015 *Communication through coherence with inter-areal delays*). In addition, the Vezoli et al Neuron 2021 study mentioned by the Reviewer also demonstrates the higher correspondence of GC to anatomical connectivity compared to other FC metrics.

Second, we specifically chose the pair-wise conditional version of Granger causality to better mitigate spurious connections that can arise when constructing functional and effective networks from time-series data. Volume conduction can result in simultaneous activity in multiple electrodes and may lead to the spurious detection of strong zero-lag (or zero-phase in the case of coherence) functional connections. Similarly, re-referencing can in some cases introduce small negative correlations between recording sites with zero time delay. The fact that Granger causality is high only when the past of one channel helps predict the future of the other (no $\tau = 0$ term in the reduced or full VAR models) helps eliminate these spurious zero-lag connections (Michalareas et al 2016 *Alpha-Beta and Gamma Rhythms Subserve Feedback and Feedforward Influences among Human Visual Cortical Areas*, Neuron). Two additional effects that can also give rise to spurious connectivity when assessing coupling via unconditioned pairwise measures are common input and transitivity. For a triplet of regions, A, B, and C, if the time delay from A to C is equal to the sum of the time delays from A to B and from B to C, then an observed connection $A \rightarrow C$ could potentially arise spuriously due to transitivity. Similarly, if the time delay from B to C is equal to the difference in time delays between A and B and A and C, then the connection $B \rightarrow C$ may potentially be present only due to a common input from A to both B and C. Pair-wise conditional Granger causality quantifies the degree to which activity of a source electrode can predict the future activity of a second target electrode over and above the degree to which the target electrode's activity is already predicted by its own past and the past of all other electrodes besides the source. Conditioning on the activity of all other recorded electrodes besides the source helps mitigate spurious causalities that may arise due to common dependencies or transitivity. We recognize that there is always the possibility that there may be sources of common drive that are not recorded by our electrodes, but our use of Granger causality mitigates this potential source of spurious connectivity as much as possible.

For these reasons, we felt that assessing connectivity in our temporal networks was most appropriate using an approach based on Granger causality. We have now provided further rationale for this choice in the revised Methods.

To further address this point and to demonstrate that our results are not simply a consequence of the metric we chose to use we have now also introduced a new analysis in which we examined connectivity in our networks using other measures of connectivity. Given the prevalence and straightforward interpretation of correlation and coherence based metrics, we computed the following for all electrode pairs: (i) raw correlation coefficient (r , zero lag), (ii) correlogram estimate of the absolute maximum cross-correlation ($|r(\tau^*)|$ where τ^* is the lag with maximum correlation), and (iii) the coherence spectrum and maximum coherence ($C(f)$ and $C(f^*)$), where f^* denotes the frequency of maximum coherence). We then used the modules that we identified (through our approach based on Granger causality) to compare these metrics between pairs that lie within modules and those that lie across different modules. We now present the results of this new analysis in Supplementary Figure S3. In brief, in individual sessions, correlations between within-module pairs are positive and of large magnitude, whereas correlations between across-module pairs are negative and of small magnitude. This was true for all individual arrays as well as across all arrays. This clear separation in correlations between within and across module pairs was also evident in individual sessions and for most arrays when using a correlogram estimate of the absolute maximum cross-correlation. Lastly, we repeated this analysis on both the coherence spectrum as well as the maximum coherence. For all sessions, the average coherence for all frequencies is larger for within-module pairs when compared to across-module pairs. Extracting the maximum coherence for each pair we find that this separation holds for most arrays and is significant across arrays. In summary, this new analysis demonstrates that while there could be quantitative differences in the identified modules depending on what FC metric is used, the metrics considered here are capturing similar information, and that the main defining feature of modularity (strong within-module connectivity vs weak across-module connectivity) holds independent of the exact measure of functional connectivity that is used.

3a. The Louvain algorithm for modularity maximisation is a nondeterministic heuristic, so it needs to be initialised with random seeds.

We completely agree with the Reviewer, and in fact we have explicitly discussed this point in our original Methods (now lines 473-482 in the revised manuscript). Because of the nondeterministic nature of the algorithm, in our original analysis we repeated the Louvain algorithm 500 times for each MEA. The data regarding the modularity for each individual array are shown in Supplementary Figures S4 and S5 in which we present the histograms that are constructed from the 500 individual runs of the Louvain algorithm. We have now clarified this point in the revised Methods.

3b. In addition, it depends on the resolution parameter gamma, which controls over the size and number of communities found (resolution equal to 1 leads to the standard Louvain method, whereas higher and lower resolutions produce larger and smaller number of clusters, respectively). It is not clear why the gamma parameter was set to 1. This is a crucial point of the study, because the major claim concerning the presence of modules of size 1.3mm strongly depends on such parameters. How can you exclude that your main claim about the spatial resolution of functional modules is not a by-product of the parameters of the analysis? What would have happened for example if the size of the MEA would have been 40mm in size rather than 4mm? We may argue that the Louvain method would have found approximately 5-7 clusters, thus giving a spatial resolution approximately 13mm (ten times as the current one). To conclude, my major concern is the indirect measure of functional modules from co-variance based temporal graphs, rather than direct neural measures.

We also agree with the Reviewer on this point that the modularity generated by the Louvain algorithm depends on the choice of parameters. As the Reviewer notes, the main results we present in our manuscript are derived using a resolution parameter γ set to 1, the standard Louvain method. However, to confirm that the choice of parameters does not significantly change our results, we have also performed a multiresolution analysis by varying γ from 0 to 2 in steps of 0.2 (example array shown in Supplementary Figure S6). We agree that the module size is a critical point of our study. We have therefore replaced the final panels in the figure to instead show the change in module size as a function γ (and ω), and have edited the corresponding supplementary text. As expected, for very small values of γ the module diameter is the size of the array since all nodes are placed into a single module. However, the module size for the real networks sharply decreases around $\gamma = 0.3-0.4$, and for values of γ from 0.8-2 the average module size remains in the range of $\sim 1-1.5$ mm. This relative stability in module size as a function of γ indicates that there is a characteristic size to the identified modules.

To further address this point and to confirm that the module sizes we identify are not artifactual, we can also leverage the fact that we have arrays of different sizes. We now show the module diameter and spatial compactness for all of the 3.2 x 3.2mm and the 4 x 4mm arrays (Supplementary Figure S5c). In general, there does not appear to be a clear systematic difference between the two array types. The diameter for one of the 4 x 4mm arrays (A2) is about the same (and sometimes smaller) than for the 3.2 x 3.2mm arrays. The modules for the other 4 x 4 mm array (A1) do have the largest diameter. However, this appears to be driven by the fact that the modules in this array are the among the least compact. We can see that the MEA with the second largest module diameter is in fact a 3.2 x 3.2 mm array (A3), and this array also has the second smallest compactness. Hence, the differences in module diameter tend to arise from differences in compactness rather than differences in array size.

4. As far as I understand, the manuscript presents results from graph theoretical analysis of Granger causality estimates computed over neural signals both during the baseline and task-related period of the visual categorisation task. If this is the case, the Granger causality estimates reflect both task-related and baseline activity. Can you explain the rationale for such a choice? Normally, task-related Granger causality displays large modulations with respect to baseline. Such differences may lead to differences in modularity analysis. Can you still claim that the modules are exclusively related to task-related activity.

We apologize for any confusion in our initial manuscript. We computed our measures of network connectivity through Granger causality by using LFP data extracted from experimental sessions during which the participants performed a variety of different cognitive testing tasks. Only a subset of these tasks were the visual categorization task, and only a subset of those tasks were ones in which we were able to capture single unit spiking activity. Single unit activity is much more challenging to capture than LFP data, and the spatial coverage is much sparser since not all microelectrodes detect single units. On the other hand, we can robustly capture clean LFP activity from most electrodes in many sessions, allowing us to compute functional connections between most pairs of electrodes across several sessions for the temporal (multi-layer) analysis of modularity. Using many time points to compute FC also increases the potential correspondence between FC and structure.

By analyzing LFP signals captured across several experimental sessions, we ensured that the participants were awake and actively engaged throughout the duration of each session.

Importantly, however, we selected the individual blocks pseudo randomly from the first 30 minutes of the session with no time-locking to any external stimulus, event, or task. Thus, the data used to extract out our measures of connectivity and modularity are captured from spontaneous, though not necessarily resting, LFP activity. Once we obtained the module membership for each electrode, we then used the available single unit activity from the categorization task to test if neurons within different modules respond differentially during cognitive processing. We believe that this correspondence between spontaneous LFP modules and task specific single unit activity strengthens our claims. We have now clarified this approach in the revised Methods, and provide information regarding the experimental sessions used in each participant to calculate both modularity and the spiking responses (Supplementary Figure S7).

5. Granger causality analysis is performed on broad-band LFPs. Current theories about brain communication through coherence, however, provide diverging hypotheses about the role of different frequency bands (alpha/beta versus gamma) in brain communication. One of the most recent studies in the field (Vezoli et al Neuron 2021, <https://www.sciencedirect.com/science/article/pii/S089662732100725X>) suggests that different frequency bands have different roles and different modular organisation. Can you please explain the rationale for taking broad-band LFP for the calculation of Granger causality?

The Reviewer raises another good point regarding the frequencies that contribute to the connectivity we measure through Granger causality. We have now addressed this concern with a new analysis. The pipeline for calculating the pair-wise conditional Granger causality used in the MVGC toolbox involves first computing the Granger causality as a function of frequency, and then integrating over all frequencies to obtain the overall Granger causality. This spectral Granger function can also be integrated over specific frequency ranges to obtain band-specific contributions to the overall Granger causality. To address the uncertainty and concern raised by the Reviewer, we have now completed new analyses in which we examine Granger causality as a function of frequency. We have included the results of these new analyses in a new main result and new main figure, Figure 3. We show that in a single example MEA, Granger causality is higher at all frequencies for within-module pairs on average compared to across module pairs. We feel that this overall difference justifies the use of the broadband Granger causality. However, if we normalize the individual Granger spectra to remove this overall shift, we can also analyze the relative contributions from different frequencies to the overall Granger causality. As a result, we find that the average frequency spectrum across sessions for within-module electrode pairs demonstrates a larger contribution from higher frequencies compared to across-module electrode pairs. To quantify this effect across MEAs we aggregated frequencies into five frequency bands: delta (2-4Hz), theta (4-8Hz), alpha (8-16Hz), beta (16-32Hz), and broadband gamma (70-150). Across all MEAs, within-module functional connectivity is mediated by significantly more high frequency activity and significantly less low frequency activity than across-module connectivity. This asymmetry is consistent with models of cortico-cortical communication which rely on oscillations at different frequencies to synchronize neuronal populations at different spatial scales. In previous work we have investigated the role of theta/alpha oscillations in large-scale cortico-cortical communication based on CTC models, and we are planning future studies to more specifically test the role of different rhythms on neural communication at the micro scale. We now raise this point in the revised Discussion.

6. Five out of the eight participants received a surgical resection which includes the tissue where the MEAs were implanted. Can you please comment on the fact that most of the recorded neural tissue was pathological and needed resection?

We appreciate the Reviewer's concern. In every one of the participants, we implanted the MEAs in the anterior temporal lobe in anticipation of performing an anterior temporal lobectomy for treatment of the participant's temporal lobe epilepsy. This is a standard surgical approach that removes the anterior 3-4 cm of the temporal lobe cortex. This resection includes regions that are clearly epileptogenic, such as more medial structures like the hippocampus, but often also includes areas of relatively normal cortex that must be removed simply to access the deeper structures. This is the case here. None of the participants had intracranial EEG recordings that suggested that the source of seizure activity was in the anterior lateral temporal cortex where the arrays were implanted, yet these regions were removed as part of the standard surgical procedure. While we cannot be certain that these regions are not pathologic in some way, the clinical recordings suggest that they are not directly involved in the seizures that were captured. Supporting this point, it is notable that we can decode category membership based on spiking activity captured from this region. This would further suggest that the tissue in this brain region does indeed encode meaningful information. We agree with the Reviewer, however, that this is an important point, and have now clarified this point in the revised Methods.

I hope the points I raised will help ameliorating the quality of the study. Best regards.

We thank the Reviewers for the very constructive and helpful comments. We have addressed the concerns raised by the Reviewers in the revised manuscript, and we agree that this has strengthened the quality of our manuscript.

Thank you again for considering our manuscript. We look forward to your reply and to the reviews of our manuscript.

Sincerely,

Kareem A. Zaghloul
Surgical Neurology Branch, NINDS
National Institutes of Health
Building 10, Room 3D20
10 Center Drive
Bethesda, MD 20892-1414
O: (301) 594-8114
F: (301) 402-0380
kareem.zaghloul@nih.gov

REVIEWERS' COMMENTS

Reviewer #1 (Remarks to the Author):

I thank the authors for addressing my comments and suggestions.

Reviewer #2 (Remarks to the Author):

In this extensive revision, the authors fully addressed the issues that I had raised. In particular, the new Fig S1 raw plots (ordered by module membership) are very helpful to see, as is the new New Fig 3 with the frequency specific GC analysis. Also, the additional metrics of within-and between cluster similarity introduced in response to Reviewer 3 are a very useful addition. I recommend publication.

Reviewer #3 (Remarks to the Author):

Dear Authors,

The reviewed manuscript has addressed in detail all my concerns. The supplementary results and figure provide novel evidence and support the main claims of the manuscript. I still have a few minor points:

1) I would suggest to consider a few more specific references in the "Spectral decomposition of conditional Granger causalities" paragraph or in the supplementary text, which make link to reviews in the domain of Granger causality (both frequency domain and conditional version, using MVAR as estimation approach) such as:

M. Ding, Y. Chen, and S. L. Bressler, "Granger causality: basic theory and application to neuroscience," in Handbook of Time Series Analysis, M. Winterhalder, B. Schelter, and J. Timmer, Eds., pp. 437–460, Wiley-VCH, Berlin, Germany, 2006.

Y. Chen, S. L. Bressler, and M. Ding, "Frequency decomposition of conditional Granger causality and application to multivariate neural field potential data," *Journal of Neuroscience Methods*, vol. 150, no. 2, pp. 228–237, 2006.

2) The legends of the novel supplementary figures are too poorly described. For example:

Figure S2: please indicate what are the colors, indicate the size of spatial scale (mm).

Figure S6: it's not clear what are the red and grey colored curves.

Congratulations for your work.

Best regards